# UDC: A Unified Neural Divide-and-Conquer Framework for Large-Scale Combinatorial Optimization Problems

**Zhi Zheng**[1]    **Changliang Zhou**[1]    **Xialiang Tong**[2]    **Mingxuan Yuan**[2]    **Zhenkun Wang**[1*]

[1] School of System Design and Intelligent Manufacturing,
Southern University of Science and Technology, Shenzhen, China
[2] Huawei Noah's Ark Lab, Shenzhen, China.
zhi.zheng@u.nus.edu, zhoucl2022@mail.sustech.edu.cn,
{tongxialiang, yuan.mingxuan}@huawei.com, wangzhenkun90@gmail.com

## Abstract

Single-stage neural combinatorial optimization solvers have achieved near-optimal results on various small-scale combinatorial optimization (CO) problems without requiring expert knowledge. However, these solvers exhibit significant performance degradation when applied to large-scale CO problems. Recently, two-stage neural methods motivated by divide-and-conquer strategies have shown efficiency in addressing large-scale CO problems. Nevertheless, the performance of these methods highly relies on problem-specific heuristics in either the dividing or the conquering procedure, which limits their applicability to general CO problems. Moreover, these methods employ separate training schemes and ignore the interdependencies between the dividing and conquering strategies, often leading to sub-optimal solutions. To tackle these drawbacks, this article develops a unified neural divide-and-conquer framework (i.e., UDC) for solving general large-scale CO problems. UDC offers a Divide-Conquer-Reunion (DCR) training method to eliminate the negative impact of a sub-optimal dividing policy. Employing a high-efficiency Graph Neural Network (GNN) for global instance dividing and a fixed-length sub-path solver for conquering divided sub-problems, the proposed UDC framework demonstrates extensive applicability, achieving superior performance in 10 representative large-scale CO problems. The code is available at `https://github.com/CIAM-Group/NCO_code/tree/main/single_objective/UDC-Large-scale-CO-master`

## 1 Introduction

Combinatorial optimization (CO) [1] has numerous practical applications including route planning [2], circuit design [3], biology [4], etc. Over the past few years, neural combinatorial optimization (NCO) methods are developed to efficiently solve typical CO problems such as the Travelling Salesman Problem (TSP) and Capacitated Vehicle Routing Problem (CVRP). Among them, reinforcement learning (RL)-based constructive NCO methods [5, 6] can generate near-optimal solutions for small-scale instances (e.g., TSP instances with no more than 200 nodes) without the requirement of expert knowledge [7]. These solvers usually adopt a single-stage solution generation process, constructing the solution of CO problems node-by-node in an end-to-end manner. However, hindered by the disability of training on larger-size instances directly [8, 9, 10], these methods exhibit limited performance when generalizing to solve large-scale CO problems.

---

[*]Corresponding author

38th Conference on Neural Information Processing Systems (NeurIPS 2024).

Table 1: Comparison between our UDC and the other existing neural divide-and-conquer methods. The proposed UDC utilizes learning-based policies in both the dividing and conquering stages. Moreover, UDC is the first to achieve a superior unified training scheme by considering the negative impact of sub-optimal dividing policies on solution generation.

| Neural Divide-and-Conquers | | Dividing Policy | Conquering Policy | Impact of Divide | Train the Two-policies |
|---|---|---|---|---|---|
| Methods | Problem | Neural✓/Heuristic✗ | | Consider✓/Ignore✗ | Unified✓/Seperate✗ |
| LCP [24] | TSP,CVRP | ✓ | ✓ | ✗ | ✗ |
| L2D [22] | CVRP | ✓ | ✗ | ✗ | ✗ |
| H-TSP [20] | TSP | ✓ | ✓ | ✗ | ✗ |
| RBG [23] | CVRP & CVRP variants | ✓ | ✓ | ✗ | ✗ |
| TAM [12] | CVRP | ✓ | ✓ | ✗ | ✗ |
| SO [21] | TSP | ✗ | ✓ | ✗ | ✗ |
| GLOP [13] | (A)TSP, CVRP, PCTSP | ✗[1] | ✓ | ✗ | ✗ |
| UDC(Ours) | General CO Problems | ✓ | ✓ | ✓ | ✓ |

To meet the demands of larger-scale CO applications, researchers have increasingly focused on extending NCO methods to larger-scale instances (e.g., 1,000-node ones) [11]. Existing NCO methods for large-scale problems primarily involve modified single-stage solvers [6] and neural divide-and-conquer methods [12, 13]. BQ-NCO [14] and LEHD [8] develop a sub-path construction process [15] and employ models with heavy-decoder to learn the process. Nevertheless, training such a heavyweight model necessitates supervised learning (SL), which limits their applicability to problems where high-quality solutions are not available as labels. ELG [6], ICAM [16], and DAR [17] integrate auxiliary information to guide the learning of RL-based single-stage constructive solvers [5]. However, the auxiliary information needs problem-specific designs, and the $\mathcal{O}(N^2)$ complexity caused by the self-attention mechanism also poses challenges for large-scale ($N$-node) problems. As another category of large-scale NCO, neural divide-and-conquer methods have attracted increasing attention. Inspired by the common divide-and-conquer idea in traditional heuristics [18, 19], these methods execute a two-stage solving process, including a dividing stage for overall instance partition and a conquering stage for sub-problem solving. By employing neural network for one or both stages, TAM [12], H-TSP [20], and GLOP [13] have demonstrated superior efficiency in large-scale TSP and CVRP.

Despite acknowledging the prospect, these neural divide-and-conquer approaches still suffer from shortcomings in applicability and solution quality. Some neural divide-and-conquer methods rely on problem-specific heuristics in either dividing (e.g., GLOP [13] and SO [21]) or conquering stages (e.g., L2D [22] and RBG [23]), which limits their generalizability across various CO problems. Moreover, all the existing neural divide-and-conquer methods adopt a separate training process, training the dividing policy after a pre-trained conquering policy [13, 12, 20]. However, the separate training scheme overlooks the interdependencies between the dividing and conquering policies and may lead to unsolvable antagonisms. Once the pre-trained conquering policy is sub-optimal for some sub-problems, such a training scheme might guide the dividing policy to adapt the pre-trained sub-optimal conquering policy, thus converging to local optima.

As shown in Table 1, this paper finds that considering the negative impact of sub-optimal dividing results is essential in generating high-quality divide-and-conquer-based solutions. To this aim, we propose a novel RL-based training method termed Divide-Conquer-Reunion (DCR). Enabling the DCR in a unified training scheme, we propose the unified neural divide-and-conquer framework (UDC) for a wide range of large-scale CO problems. UDC employs a fast and lightweight GNN to decompose a large-scale CO instance into several fixed-length sub-problems and utilizes constructive solvers to solve these sub-problems. We conduct extensive experiments on **10** different CO problems. Experimental results indicate that UDC can significantly outperform existing large-scale NCO solvers in both efficiency and applicability without relying on any heuristic design.

The contributions of this paper are as follows: **1)** We propose a novel method DCR to enhance training by alleviating the negative impact of sub-optimal dividing policies. **2)** Leveraging DCR in training, the proposed UDC achieves a unified training scheme with significantly superior performance. **3)** UDC demonstrates extensive applicability and can be applied to general CO problems with similar settings.

---

[1]GLOP [13] utilizes heuristic dividing policy for TSP and Asymmetric TSP (ATSP) and neural dividing policy for CVRP and Prize Collecting TSP (PCTSP).

## 2 Preliminaries: Neural Divide-and-Conquer

### 2.1 Divide-and-Conquer

A CO problem with $N$ decision variables is defined to minimize the objective function $f$ of a solution $\boldsymbol{x} = (x_1, x_2, \ldots, x_N)$ as follows:

$$\underset{\boldsymbol{x} \in \Omega}{\text{minimize}} \quad f(\boldsymbol{x}, \mathcal{G}) \tag{1}$$

where $\mathcal{G}$ represents the CO instance and $\Omega$ is a set consisting of all feasible solutions.

Leveraging the divide-and-conquer idea in solving CO problems is promising in traditional (meta)heuristic algorithms [25], especially large-neighborhood-search-based algorithms [26, 27]. These methods achieve gradual performance improvement from an initial solution through iterative sub-problem selection (i.e., the dividing stage) and sub-problem repair (i.e., the conquering stage). Recent neural divide-and-conquer methods conduct a similar solving process and employ efficient deep-learning techniques to model the policies of both stages [13]. The dividing policy $\pi_d(\mathcal{G})$ decomposes the instance $\mathcal{G}$ to $K$ mutually unaffected sub-problems $\{\mathcal{G}_1, \mathcal{G}_2, \ldots, \mathcal{G}_K\}$ and the conquering policy $\pi_c$ generates sub-solutions $\boldsymbol{s}_k$ for $k \in \{1, \ldots, K\}$ by $\boldsymbol{s}_k \sim \pi_c(\mathcal{G}_k)$ to minimize the objective function $f'(\boldsymbol{s}_k, \mathcal{G}_k)$ of corresponding sub-problems. Finally, a total solution $\boldsymbol{x}$ of $\mathcal{G}$ is merged as $\boldsymbol{x} = \text{Concat}(\boldsymbol{s}_1, \ldots, \boldsymbol{s}_K)$.

### 2.2 Constructive Neural Solver

The (single-stage) constructive NCO solvers are promising in solving general small-scale CO problems with time efficiency. These constructive solvers typically employ an attention-based encoder-decoder network structure, where a multi-layer encoder processes CO instances into embeddings in hidden spaces and a lightweight decoder handles dynamic constraints while constructing feasible solutions [5]. With the solving process modeled as Markov Decision Processes (MDPs) [28], constructive solvers can be trained using deep reinforcement learning (DRL) techniques without requiring expert experience. In the solution generation process, constructive solvers with parameter $\theta$ process the policy $\pi$ of a $\tau$-length solution $\boldsymbol{x} = (x_1, \ldots, x_\tau)$ as follows:

$$\pi(\boldsymbol{x}|\mathcal{G}, \Omega, \theta) = \prod_{t=1}^{\tau} p_\theta(x_t|\boldsymbol{x}_{1:t-1}, \mathcal{G}, \Omega), \tag{2}$$

where $\boldsymbol{x}_{1:t-1}$ represents the partial solution before the selection of $x_t$. DRL-based constructive NCO solvers demonstrate outstanding solution qualities in small-scale CO problems. So existing neural divide-and-conquer methods [13, 21] also generally employ a pre-trained constructive neural solver (e.g., Attention Model [28]) for their conquering policy $\pi_c$.

### 2.3 Heatmap-based Neural Solver

Advanced constructive NCO solvers rely on the self-attention layers [29] for node representation, whose quadratic time and space complexity related to problem scales [30] hinders the direct training on large-scale instances. So, heatmap-based solvers [31, 32] are proposed to solve large-scale CO problems efficiently by a lighter-weight GNN [33]. In heatmap-based solvers for vehicle routing problems (VRPs), a GNN with parameter $\phi$ first generates a heatmap $\mathcal{H} \in \mathbb{R}^{N \times N}$ within linear time [34] and a feasible solution is then constructed based on this heatmap with the policy [35] as follows:

$$\pi(\boldsymbol{x}|\mathcal{G}, \Omega, \theta) = p_\theta(\mathcal{H}|\mathcal{G}, \Omega)p(x_1) \prod_{t=2}^{\tau} \frac{exp(\mathcal{H}_{x_{t-1}, x_t})}{\sum_{i=t}^{N} exp(\mathcal{H}_{x_{t-1}, x_i})}, \tag{3}$$

However, the non-autoregressively generated heatmap [36] excludes the information about the order of the constructed partial solution $\boldsymbol{x}_{1:t-1}$, so the greedy decoding policy in heatmap-based solvers becomes low-quality and these solvers rely on search algorithms [37, 38] for higher-quality solutions.

## 3 Methodology: Unified Divide-and-Conquer

As shown in Figure 1, UDC follows the general framework of neural divide-and-conquer methods [13], solving CO problems through two distinct stages: the dividing stage and the conquering stage. The

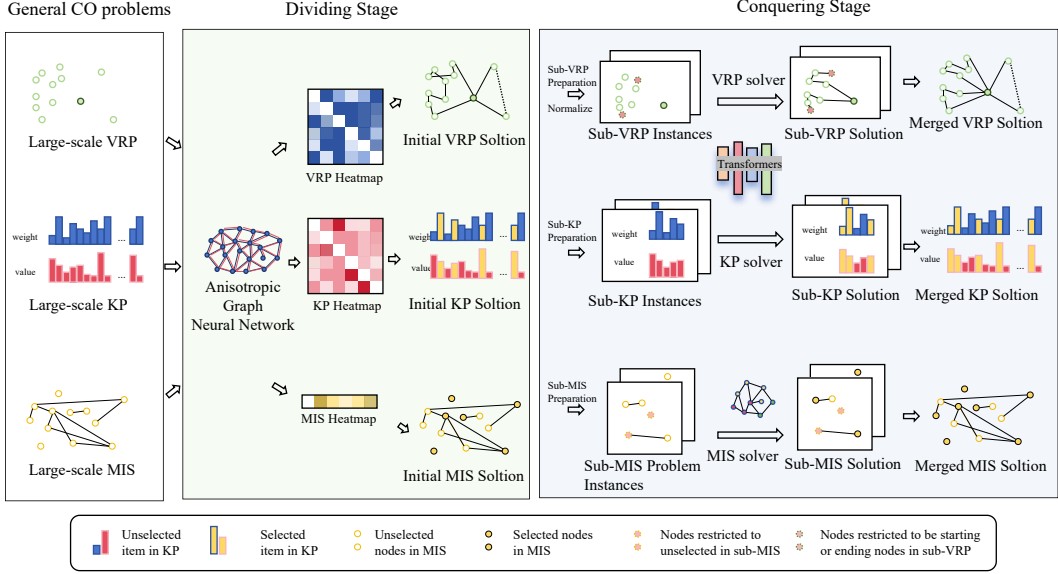

Figure 1: The solving process of the proposed UDC on large-scale VRP, 0-1 Knapsack Problem (KP), and Maximum Independent Set (MIS).

dividing stage of UDC is expected to generate a feasible initial solution with a similar overall shape compared to the optimal solution, so the subsequent conquering stage can obtain more high-quality solutions by solving local sub-problems. For all the involved CO problems, UDC first constructs a sparse graph to represent the instance and then employs a heatmap-based solver with Anisotropic Graph Neural Network (AGNN) as the dividing policy [39, 40] to generate initial solutions. The following conquering stage first decomposes the original CO instance into multiple sub-problems based on the initial solution while integrating the necessary constraints for sub-problems. Then, towards minimizing the decomposed objective function of the sub-problem, UDC uses constructive neural solvers as the conquering policy to generate sub-solutions. There is no single universal constructive neural solver that can handle all types of CO problems [41, 42], so we employ suitable solvers for each sub-CO-problem (e.g., AGNN [39] for MIS, POMO [5] or ICAM [16] for VRPs).

## 3.1 Pipeline: Dividing Stage & Conquering Stage

**Dividing Stage: Heatmap-based Neural Dividing:** Heatmap-based solvers require less time and space consumption compared to constructive solvers, so they are more suitable for learning global dividing policies on large-scale instances. The dividing stage first constructs a sparse graph $\mathcal{G}_D = \{\mathbb{V}, \mathbb{E}\}$ (i.e., linking K-nearest neighbors (KNN) in TSP, or original edges in $\mathcal{G}$ for MIS) for the original CO instance $\mathcal{G}$. Then, we utilize Anisotropic Graph Neural Networks (AGNN) with parameter $\phi$ to generate the heatmap $\mathcal{H}$. For $N$-node VRPs, the heatmap $\mathcal{H} \in \mathbb{R}^{N \times N}$ and a $\tau$-length initial solutions $\boldsymbol{x}_0 = (x_{0,1}, \ldots, x_{0,\tau})$ is generated based on the policy $\pi_d$ as follows:

$$\pi_d(\boldsymbol{x}_0|\mathcal{G}_D, \Omega, \phi) = \begin{cases} p(\mathcal{H}|\mathcal{G}_D, \Omega, \phi)p(x_{0,1}) \prod_{t=2}^{\tau} \dfrac{\exp(\mathcal{H}_{x_{0,t-1},x_{0,t}})}{\sum_{i=t}^{N} \exp(\mathcal{H}_{x_{0,t-1},x_{0,i}})}, & \text{if } \boldsymbol{x}_0 \in \Omega \\ 0, & \text{otherwise} \end{cases}. \quad (4)$$

**Conquering Stage: Sub-problem Preparation:** The conquering stage first generates pending sub-problems along with their specific constraints. For VRPs, the nodes in sub-VRPs and the constraints of sub-VRPs are built based on continuous sub-sequences in the initial solution $\boldsymbol{x}_0$. In generating $n$-node sub-problems, UDC divides the original $N$-node instance $\mathcal{G}$ into $\lfloor \frac{N}{n} \rfloor$ sub-problems $\{\mathcal{G}_1, \ldots, \mathcal{G}_{\lfloor \frac{N}{n} \rfloor}\}$ according to $\boldsymbol{x}^0$, temporarily excluding sub-problems with fewer than $n$ nodes. The constraints $\{\Omega_1, \ldots, \Omega_{\lfloor \frac{N}{n} \rfloor}\}$ of sub-problems include not only the problem-specific constraints (e.g., no self-loop in sub-TSPs) but also additional constraints to ensure the legality of the merged solution after

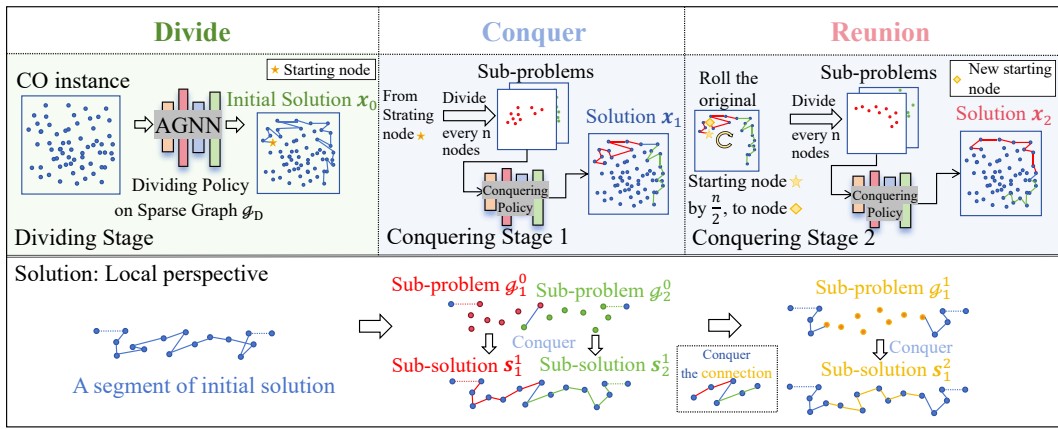

Figure 2: The proposed DCR training method. The upper part shows the pipeline of a single DCR-enabled training step, including a Divide step, a Conquer step, and a Reunion step. The lower part provides a local view of a solution fragment to demonstrate our motivation, the additional Reunion step can repair the wrong connections from sub-optimal sub-problem dividing results. DCR has the potential to correct sub-optimal dividing results by repairing the connection part in the additional Reunion step.

integrating the sub-problem solution to the original solution (e.g., maintaining the first and last node in sub-VRPs). Similar to previous works [13, 43], we also normalize the coordinate of sub-problems and some numerical constraints to enhance the homogeneity of pending sub-problem instances. Due to the need to handle different constraints, the sub-problem preparation processes vary for different CO problems, which is presented in detail in the Appendix B.

**Conquering Stage: Constructive Neural Conquering:** After the preparation of sub-problems, the conquering policy is utilized to generate solutions to these instances $\{\mathcal{G}_1, \ldots, \mathcal{G}_{\lfloor \frac{N}{n} \rfloor}\}$. We utilize constructive solvers with parameter $\theta$ for most involved sub-CO problems. Their sub-solution $\boldsymbol{s}_k = (s_{k,1}, \ldots, s_{k,n}), k \in \{1, \ldots, \lfloor \frac{N}{n} \rfloor\}$ are sampled from the conquering policy $\pi_c$ as follows:

$$\pi_c(\boldsymbol{s}_k|\mathcal{G}_k, \Omega_k, \theta) = \begin{cases} \prod_{t=1}^{n} p(\boldsymbol{s}_{k,t}|\boldsymbol{s}_{k,1:t-1}\mathcal{G}_k, \Omega_k, \theta), & \text{if } \boldsymbol{s}_k \in \Omega_k \\ 0, & \text{otherwise} \end{cases}, \tag{5}$$

where $\boldsymbol{s}_{k,1:t-1}$ represents the partial solution before the selection of $s_{k,t}$. Finally, as the last step in a conquering stage, sub-solutions with improvements on the objective function will replace the original solution fragment in $\boldsymbol{x}_0$ and the merged solution becomes $\boldsymbol{x}_1$. It's worth noting that the conquering stage can execute repeatedly on the new merged solution for a gradually better solution quality and we note the solution after $r$ conquering stages as $\boldsymbol{x}_r$.

## 3.2 Training Method: Divide-Conquer-Reunion (DCR)

Both the dividing and conquering stages in neural divide-and-conquer methods can be modeled as MDPs [12] (shown in Appendix C.7). The reward for the conquering policy is derived from the objective function of the sub-solutions, while the reward for the dividing policies is calculated based on the objective function of the final merged solution. Existing neural divide-and-conquer methods cannot train both policies simultaneously with RL [20]. Therefore, they generally employ a separate training scheme, pre-training a conquering policy on specific datasets before training the dividing policy [13]. However, this separate training process not only requires additional problem-specific dataset designs [13] but also undermines the collaborative optimization of the dividing and conquering policies (further discussed in Section 5.1).

In this paper, we imply that considering the negative impact of sub-optimal sub-problems decomposition is essential to implement the unified training of both policies. Sub-problems generated by sub-optimal dividing policies will probably not match any continuous segments in the optimal

solution so as shown in Figure 2, despite obtaining high-quality sub-solutions, there may be error connections emerging at the linking regions of these sub-problems. The naive solving procedure of neural divide-and-conquer methods ignores these inevitable errors in solution generation, thus utilizing biased rewards to assess the dividing policy. To tackle this drawback, we propose the Divide-Conquer-Reunion (DCR) process to eliminate the impact as much as possible. DCR designs an additional Reunion step to treat the connection between the original two adjacent sub-problems as a new sub-problem and conduct another conquering stage on $x_1$ with new sub-problem decompositions. The Reunion step can be formulated as rolling the original start point $x_{1,0}$ along the solution $x_{1,0}$ by $l$ times (i.e., to $x_{1,l}$) and conquering again. To ensure equal focus on the adjacent two sub-problems, we set $l = \frac{n}{2}$ in training all the DCR-enabled UDC models. DCR provides a better estimation for the reward of dividing policies, thereby increasing the stability and convergence speed in unified training.

UDC adopts the REINFORCE algorithm [44] in training both the dividing policy (with Loss $\mathcal{L}_d$) and the conquering policy (with Loss $\mathcal{L}_{c1}$ for the Conquer step and $\mathcal{L}_{c2}$ for the Reunion step). The baseline in training dividing policy is the average reward over $\alpha$ initial solutions $x_0^i, i \in \{1, \ldots, \alpha\}$ and the baseline for conquering policy is calculated over $\beta$ sampled sub-solutions. The gradients of loss functions on a single instance $\mathcal{G}$ are calculated as follows:

$$\nabla \mathcal{L}_d(\mathcal{G}) = \frac{1}{\alpha} \sum_{i=1}^{\alpha} \left[ \left(f(x_2^i, \mathcal{G}) - \frac{1}{\alpha} \sum_{j=1}^{\alpha} f(x_2^j, \mathcal{G})\right) \nabla \ \log \pi_d(x_2^i | \mathcal{G}_D, \Omega, \phi) \right],$$

$$\nabla \mathcal{L}_{c1}(\mathcal{G}) = \frac{1}{\alpha \beta \lfloor \frac{N}{n} \rfloor} \sum_{c=1}^{\alpha \lfloor \frac{N}{n} \rfloor} \sum_{i=1}^{\beta} \left[ \left(f'(s_c^{1,i}, \mathcal{G}_c^0) - \frac{1}{\beta} \sum_{j=1}^{\beta} f'(s_c^{1,j}, \mathcal{G}_c^0)\right) \nabla \ \log \pi_c(s_c^{1,j} | \mathcal{G}_c^0, \Omega_c^0, \theta) \right],$$

$$\nabla \mathcal{L}_{c2}(\mathcal{G}) = \frac{1}{\alpha \beta \lfloor \frac{N}{n} \rfloor} \sum_{c=1}^{\alpha \lfloor \frac{N}{n} \rfloor} \sum_{i=1}^{\beta} \left[ \left(f'(s_c^{2,i}, \mathcal{G}_c^1) - \frac{1}{\beta} \sum_{j=1}^{\beta} f'(s_c^{2,j}, \mathcal{G}_c^1)\right) \nabla \ \log \pi_c(s_c^{2,j} | \mathcal{G}_c^1, \Omega_c^1, \theta) \right],$$
(6)

where $\{x_2^1, \ldots, x_2^\alpha\}$ represents the $\alpha$ sampled solutios. There are $\alpha \lfloor \frac{N}{n} \rfloor$ sub-problems $\mathcal{G}_c^0, c \in \{1, \ldots, \lfloor \frac{N}{n} \rfloor, \ldots, \alpha \lfloor \frac{N}{n} \rfloor\}$ generated based on $\{x_0^1, \ldots, x_0^\alpha\}$ in the first conquering stage with constraints $\Omega_c^0$. $\alpha \lfloor \frac{N}{n} \rfloor$ can be regarded as the batch size of sub-problems, and $\mathcal{G}_c^1, \Omega_c^1, c \in \{1, \ldots, \alpha \lfloor \frac{N}{n} \rfloor\}$ is the sub-problems and constraints in the second conquering stage. The $\beta$ sampled sub-solutions for sub-problem $\mathcal{G}_c^0, \mathcal{G}_c^1, c \in \{1, \ldots, \alpha \lfloor \frac{N}{n} \rfloor\}$ are noted as $\{s_c^{1,i}, \ldots, s_c^{1,\beta}\}, \{s_c^{2,i}, \ldots, s_c^{2,\beta}\}$.

### 3.3 Application: Applicability in General CO Problems

This subsection discusses the applicability of UDC to various CO problems. Most existing neural divide-and-conquer methods rely on problem-specific designs, so they are only applicable to limited CO problems. UDC uses no heuristics algorithms or problem-specific designs in the whole solving process, thus can be applied to general offline CO problems that satisfy the following three conditions:

- The objective function contains only decomposable aggregate functions (i.e., no functions like Rank or Top-k).
- The legality of initial solutions and sub-solutions can be ensured with feasibility masks.
- The solution of the divided sub-problem is not always unique.

For the second condition, the proposed UDC is disabled on complex CO problems whose solution cannot be guaranteed to be legal through an autoregressive solution process (e.g., TSPTW). For the third condition, on certain CO problems such as the MIS problem on dense graphs or job scheduling problems, the solution of sub-problems has already been uniquely determined, so the conquering stages become ineffective.

## 4 Experiment

To verify the applicability of UDC in the general CO problem, we evaluate the UDC on 10 combinatorial optimization problems that satisfy the conditions proposed in Section 3.3, including TSP, CVRP, KP, MIS, ATSP, Orienteering Problem (OP), PCTSP, Stochastic PCTSP (SPCTSP), Open

Table 2: Objective function (Obj.), Gap to the best algorithm (Gap), and solving time (Time) on 500-node. 1,000-node, and 2,000-node TSP and CVRP. All TSP test sets and CVRP500 test sets contain 128 instances. CVRP1,000 and CVRP2,000 contain 100 instances (following the generation settings in [13]). The overall best performance is in bold and the best learning-based method is marked by shade.

| | Methods | $N=500$ | | | $N=1,000$ | | | $N=2,000$ | | |
| --- | --- | --- | --- | --- | --- | --- | --- | --- | --- | --- |
| | | Obj.↓ | Gap | Time | Obj.↓ | Gap | Time | Obj.↓ | Gap | Time |
| TSP | LKH3 | **16.52** | - | 5.5m | **23.12** | - | 24m | **32.45** | - | 1h |
| | Attn-MCTS* | 16.97 | 2.69% | 6m | 23.86 | 3.20% | 12m | 33.42 | 2.99% | - |
| | DIMES* | 16.84 | 1.93% | 2.2h | 23.69 | 2.47% | 4.6h | - | - | - |
| | DIFUSCO+Search* | 17.48 | 5.80% | 19m | 25.11 | 8.61% | 59.2m | - | - | - |
| | BQ | 16.72 | 1.18% | 46s | 23.65 | 2.27% | 1.9m | 34.03 | 4.86% | 3m |
| | LEHD | 16.78 | 1.56% | 16s | 23.85 | 3.17% | 1.6m | 34.71 | 6.98% | 12m |
| | POMO | 20.19 | 22.19% | 1m | 32.50 | 40.57% | 8m | 50.89 | 56.85% | 10m |
| | ELG | 17.18 | 4.00% | 2.2m | 24.78 | 7.18% | 13.7m | 36.14 | 11.37% | 14.2m |
| | ICAM | 16.65 | 0.77% | 38s | 23.49 | 1.58% | 1m | 34.37 | 5.93% | 3.8m |
| | GLOP(more revisions) | 16.88 | 2.19% | 1.5m | 23.84 | 3.12% | 3.5m | 33.71 | 3.89% | 9m |
| | SO-mixed* | 16.94 | 2.55% | 32m | 23.77 | 2.79% | 32m | 33.35 | 2.76% | 14m |
| | H-TSP | 17.55 | 6.22% | 20s | 24.72 | 6.91% | 40s | 34.88 | 7.49% | 48s |
| | UDC-$x_2$($\alpha$=50, greedy) | 16.94 | 2.53% | 20s | 23.79 | 2.92% | 32s | 34.14 | 5.19% | 23s |
| | UDC-$x_{50}$($\alpha$=50) | 16.78 | 1.58% | 4m | 23.53 | 1.78% | 8m | 33.26 | 2.49% | 15m |
| CVRP | LKH3 | **37.23** | - | 7.1h | 46.40 | 7.90% | 14m | 64.90 | 8.14% | 40m |
| | BQ | 38.44 | 3.25% | 47s | 44.17 | 2.72% | 55s | 62.59 | 4.29% | 3m |
| | LEHD | 38.41 | 3.18% | 17s | 43.96 | 2.24% | 1.3m | 61.58 | 2.62% | 9.5m |
| | ELG | 38.34 | 2.99% | 2.6m | 43.58 | 1.33% | 15.6m | - | - | - |
| | ICAM | 37.49 | 0.69% | 42s | 43.07 | 0.16% | 26s | 61.34 | 2.21% | 3.7m |
| | L2D* | - | - | - | 46.3 | 7.67% | 3m | 65.2 | 8.64% | 1.4h |
| | TAM(LKH3)* | - | - | - | 46.3 | 7.67% | 4m | 64.8 | 7.97% | 9.6m |
| | GLOP-G (LKH-3) | 42.45 | 11.73% | 2.4m | 45.90 | 6.74% | 2m | 63.02 | 5.01% | 2.5m |
| | UDC-$x_2$($\alpha$=50, greedy) | 40.04 | 7.57% | 1m | 46.09 | 7.18% | 2.1m | 65.26 | 8.75% | 5m |
| | UDC-$x_{50}$($\alpha$=50) | 38.34 | 3.01% | 7.7m | 43.48 | 1.11% | 14m | 60.94 | 1.55% | 23m |
| | UDC-$x_{250}$($\alpha$=50) | 37.99 | 2.06% | 35m | **43.00** | - | 1.2h | **60.01** | - | 2.15h |

VRP (OVRP), and min-max multi-agent TSP (min-max mTSP). Detailed formulations of these problems are provided in the Appendix B.

**Implementation** By employing lightweight networks for dividing policies, UDC can be directly trained on large-scale CO instances (e.g., TSP500 and TSP1,000). To learn more scale-independent features, we follow the varying-size training scheme [45, 46] in training and train only one UDC model for each CO problem. The Adam optimizer [47] is used for all involved CO problems and the detailed hyperparameters for each CO problem are provided in Appendix D.1. Most training tasks can be completed within 10 days on a single NVIDIA Tesla V100S GPU.

**Baselines** This paper compares the proposed UDC to advanced heuristic algorithms and existing neural solvers for large-scale CO problems, including **classical solvers** (LKH [48], EA4OP [49], OR-Tools [50]), **single-stage SL-based sub-path solvers** (BQ [14], LEHD [8]), **single-stage RL-based constructive solvers** (AM [51], POMO[5], ELG [6], ICAM [16], MDAM [52], et al.), **heatmap-based solvers** (DIMES [35], DIFUSCO [39], et al.) and **neural divide-and-conquer methods** (GLOP [13], TAM [12], H-TSP[20], et al.). The implementation details and settings of all the baseline methods are described in Appendix G.

**Results on 500-node to 2,000-node Instances.** We first examine the performance of UDC models on large-scale CO problems ranging from 500-node to 2,000-node. Both single-stage SL-based constructive methods [16] and single-stage sub-path solvers [8] cannot be directly trained on this scale, so we report the results of models trained on the maximum available scale for these baselines. For UDC, we provide not only the greedy results (UDC-$x_2$) but also the results after conducting more conquering stages (i.e., 50 and 250 stages, labeled as UDC-$x_{50}$, UDC-$x_{250}$). In the following conquering stages after a greedy solution, the starting point of sub-problem dividing is no longer fixed at $\frac{n}{2}$ but is sampled from Uniform$(0, n)$. For better performances, we sample $\alpha = 50$ initial

Table 3: Experiment results on 500-node to 2,000-node OP and PCTSP and SPCTSP. The 500-scale, 100-scale, and 2,000-scale problems contain 128, 128, and 16 instances, respectively. "bs-$k$" represents $k$-width beam search [51]. The overall best performance is in bold and the best learning-based method is marked by shade.

| | Methods | $N=500$ | | | $N=1,000$ | | | $N=2,000$ | | |
| | | Obj.↑[2] | Gap | Time | Obj.↓[2] | Gap | Time | Obj.↓[2] | Gap | Time |
|---|---|---|---|---|---|---|---|---|---|---|
| **OP (dist)** | EA4OP | **81.70** | - | 7.4m | **117.70** | - | 63.4m | **167.74** | - | 38.4m |
| | BQ-bs16* | - | 4.10% | 10m | - | 10.68% | 17m | - | - | - |
| | AM | 64.45 | 21.11% | 2s | 79.28 | 32.64% | 5s | 99.37 | 40.76% | 2s |
| | AM-bs1024 | 68.59 | 16.04% | 1.2m | 79.65 | 32.32% | 4m | 95.48 | 43.08% | 1.6m |
| | MDAM-bs50 | 65.78 | 19.49% | 2.6m | 81.35 | 30.88% | 12m | 105.43 | 37.15% | 7m |
| | Sym-NCO | 73.22 | 10.38% | 27s | 95.36 | 18.98% | 1.5m | 121.37 | 27.64% | 20s |
| | UDC-$x_2(\alpha=1)$ | 75.28 | 7.86% | 17s | 109.42 | 7.03% | 25s | 135.34 | 19.32% | 5s |
| | UDC-$x_{50}(\alpha=1)$ | 78.02 | 4.51% | 23s | 111.72 | 5.08% | 33s | 139.25 | 16.99% | 8s |
| | UDC-$x_{250}(\alpha=1)$ | 79.27 | 2.98% | 48s | 113.27 | 3.76% | 1.1m | 144.10 | 14.09% | 21s |
| **PCTSP** | OR-Tools* | 14.40 | 9.38% | 16h | 20.60 | 12.66% | 16h | - | - | - |
| | AM-bs1024 | 19.37 | 47.13% | 14m | 34.82 | 90.42% | 23m | 87.53 | 249.60% | 10m |
| | MDAM-bs50 | 14.80 | 13.30% | 6.5m | 22.21 | 21.43% | 53m | 36.63 | 46.30% | 43m |
| | Sym-NCO | 19.46 | 47.81% | 1.3m | 29.44 | 61.01% | 7m | 36.95 | 47.58% | 2m |
| | GLOP | 14.30 | 9.03% | 1.5m | 19.80 | 8.08% | 2.5m | 27.25 | 8.84% | 1m |
| | UDC-$x_2(\alpha=1)$ | 13.95 | 5.99% | 34s | 19.50 | 6.65% | 1m | 27.22 | 8.73% | 18s |
| | UDC-$x_{50}(\alpha=1)$ | 13.25 | 0.64% | 42s | 18.46 | 0.97% | 1.3m | 25.46 | 1.68% | 23s |
| | UDC-$x_{250}(\alpha=1)$ | **13.17** | - | 80s | **18.29** | - | 1.5m | **25.04** | - | 53s |
| **SPCTSP** | AM-bs1024 | 33.91 | 156.11% | 1.2m | 47.55 | 156.42% | 4m | 76.02 | 199.48% | 14m |
| | MDAM | 14.80 | 11.80% | 6.5m | 21.96 | 18.42% | 53m | 35.15 | 38.48% | 43m |
| | UDC-$x_2(\alpha=1)$ | 14.21 | 7.32% | 42s | 20.01 | 7.92% | 1.1m | 28.15 | 10.91% | 20s |
| | UDC-$x_{50}(\alpha=1)$ | 13.40 | 1.12% | 1m | 18.93 | 2.06% | 1.4m | 26.24 | 3.38% | 29s |
| | UDC-$x_{250}(\alpha=1)$ | **13.24** | - | 2.3m | **18.54** | - | 2.6m | **25.38** | - | 1.1m |

solutions $x_0$ for TSP and CVRP. The comparison results of TSP and CVRP are shown in Table 2 and the results of OP, PCTSP, and SPCTSP are exhibited in Tabel 4, UDC demonstrates the general applicability, exhibiting consistent competitive performances in all the five involved CO problems. After multiple conquering stages, UDC can achieve significantly better results.

UDC variants demonstrate the best results in PCTSP, SPCTSP, CVRP1,000, and CVRP2,000, together with the best learning-based method in most involved CO problems. The UDC-$x_{250}(\alpha=1)$ showcases extensive efficiency in OP, PCTSP, and PCTSP. Compared to sub-path solvers BQ [14] and LEHD [8], UDC variants exhibit more competitive results on TSP1,000, TSP2,000, CVRP, and OP. RL-based constructive solver ICAM [16] with 8 augmentation [5] is outstanding performance on TSP500, TSP1,000, and CVRP500 (i.e., relatively smaller scale) but UDC variants demonstrates better effect and efficiency in other test sets. Moreover, compared to existing neural divide-and-conquer methods GLOP [13], H-TSP [20], and TAM(LKH3) [12], UDC achieves significant advantages in performance with no heuristic design, which preliminarily implies the significance of a unified training scheme for neural divide-and-conquer methods.

**Results on Benchmark Datasets.** We also evaluate the proposed UDC on benchmark datasets TSPLib [53] and CVRPLib [54, 55]. We adopt the instances with 500-node to 5,000-node in TSPLib, CVRP Set-X [54], and very large-scale CVRP dataset Set-XXL as test sets. Instances in these datasets often with special distributions and can better represent real-world applications. As shown in Table 4, UDC variant UDC-$x_{250}$ exhibits competitive benchmark results, demonstrating the best learning-based result on CVRPLib. The proposed UDC can also demonstrate superiority in the other 5 involved CO problems, very large-scale instances. Results of these experiments are shown in Appendix D.2 and Appendix D.5, respectively. And as to AppendixD.3 and AppendixD.4, UDC can also exhibit outstanding out-of-domain generalization ability on TSP, CVRP, and MIS instances with different distributions.

---

[2]OP aims to maximize the objective function, while PCTSP and SPCTSP aim to minimize the objective function.

Table 4: TSPLib and CVRPLib results, the best learning-based result is marked by shade

| Dataset, $N \in$ | LEHD | ELG aug×8 | GLOP-LKH3 | TAM(LKH3) | UDC-$\boldsymbol{x}_2$ | UDC-$\boldsymbol{x}_{250}$ |
|---|---|---|---|---|---|---|
| TSPLib, Gap to Best Known Solution | | | | | | |
| TSPLib,(500,1,000] | 4.1% | 8.7% | 4.0% | - | 9.5% | 6.0% |
| TSPLib,(1,000,5,000] | 11.3% | 15.5% | 6.9% | - | 12.8% | 7.6% |
| CVRPLib, Gap to Best Known Solution | | | | | | |
| Set-X,(500,1,000] | 17.4% | 7.8% | 16.8% | 9.9% | 16.5% | 7.1% |
| Set-XXL,(1,000,10,000] | 22.2% | 15.2% | 19.1% | 20.4% | 31.3% | 13.2% |

## 5 Discussion

The experimental results have preliminarily proven that the proposed UDC method achieves excellent results on a wide range of large-scale CO problems with outstanding efficiency. In this section, we conduct ablation experiments to verify the necessity of components in training UDC models. Appendix E includes the ablation experiments on the testing procedure of UDC, including the ablation on the number of conquering stages $r$ and the number of sampled solutions $\alpha$.

### 5.1 Unified Training Scheme versus Separate Training Scheme

For neural divide-and-conquer methods, the unified training process does not require special designs for pre-training conquering policies, thus being easier to implement compared to separate training processes. Moreover, unified training can also avoid the antagonism between the dividing and conquering policies. To validate this, we pre-train a conquering policy (i.e., ICAM) on uniform TSP100 data and then train two ablation variants of the original UDC (separate training scheme): one variant uses the pre-trained conquering policy in the subsequent joint training (i.e., Pre-train + Joint Training, similar to H-TSP [20]), and the other variant only train the dividing policy afterward (i.e., Pre-train + Train Dividing, similar to TAM [12] & GLOP [13]). According to the training curves of UDC and the two variants in Figure 3, the unified training scheme demonstrates superiority, while the Pre-train step leads the training into local optima, which significantly harms the convergence of UDC.

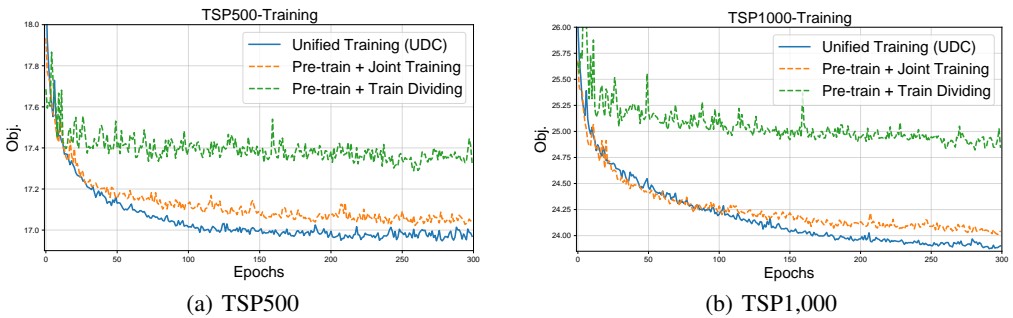

(a) TSP500      (b) TSP1,000

Figure 3: Training curves of UDC variants with different training schemes.

### 5.2 Ablation study: Conquering Policy

UDC adopts the state-of-the-art constructive solver ICAM [16] for conquering sub-TSP. However, the selection of constructive solvers for the conquering policy is not limited. We employ POMO [33] for the conquering policy, with experiment results of this version (i.e., UDC(POMO)) on TSP500 and TSP1,000 shown in Table 5.1 ($\alpha = 50$ for all variants). The UDC framework demonstrates conquering-policy-agnosticism, as it does not show significant performance degradation when the conquering policy becomes POMO.

Table 5: Results of UDC variants with various conquering policies.

| Optimality Gap | TSP500 | TSP1,000 |
|---|---|---|
| UDC(ICAM)-$\boldsymbol{x}_2$ | 2.54% | 2.92% |
| UDC(ICAM)-$\boldsymbol{x}_{50}$ | 1.58% | 1.78% |
| UDC(POMO)-$\boldsymbol{x}_2$ | 2.64% | 3.41% |
| UDC(POMO)-$\boldsymbol{x}_{50}$ | 1.64% | 1.82% |

In Appendix E.4, we further validate the conquering-policy-agnosticism of UDC on KP. We also conduct other ablation studies to demonstrate the necessity of other components of the proposed UDC, including the Reunion step in the DCR training method and the coordinate transformation in the conquering stage.

## 6   Conclusion, Limitation, and Future Work

This paper focuses on neural divide-and-conquer solvers and develops a novel training method DCR for neural divide-and-conquer methods by alleviating the negative impact of sub-optimal dividing results. By using DCR in training, this paper enables a superior unified training scheme and proposes a novel unified neural divide-and-conquer framework UDC. UDC exhibits not only outstanding performances but also extensive applicability, achieving significant performance advantages in 10 representative large-scale CO problems.

**Limitation and Future Work.** Although UDC has achieved outstanding performance improvements, we believe that UDC can achieve better results through designing better loss functions. In the future, we will focus on a more suitable loss for the UDC framework to further improve training efficiency. Moreover, we will attempt to extend the applicability of UDC to CO problems mentioned in Section 3.3 that are not applicable in the current version.

## Acknowledge

This work was supported by the National Natural Science Foundation of China (Grant No. 62106096 and Grant No. 62476118), the Natural Science Foundation of Guangdong Province (Grant No. 2024A1515011759), the National Natural Science Foundation of Shenzhen (Grant No. JCYJ20220530113013031).

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

# Appendix

In the following Appendix, we will provide a literature review A, detailed definitions of CO problems & sub-CO problems B, details of the proposed UDC method C, additional experimental results D, additional ablation study E, and visualizations on a TSP1,000 instance, a CVRP1,000 instance (i.e., X-n1001-k43 in CVRPLib), and a OP1,000 instance F.

## A  Related Work

### A.1  Neural Combinatorial Optimization

Over the past few years, extensive research on learning-based methods has been applied to various CO problems. Pointer Network [56] first applies the deep learning technique to solve NP-hard CO problems, after that, [28] designs an RL-based framework for training NCO solvers. In recent years, more diverse and sophisticated NCO frameworks have been proposed [57, 58]. Among them, the mainstream methods include (RL-based) constructive solvers [51], (RL-based) improvement-based solvers [59, 60], (SL-based) sub-path solvers [8], the heatmap-based solvers [35], and the neural divide-and-conquer methods [13]. These categories of methods demonstrate diverse advantages across different scales and CO problems.

### A.2  Constructive Neural Solvers

The basic idea of constructive neural solvers has been illustrated in Section 2.2. Up to now, constructive neural solvers are the most popular neural method for solving CO problems. By employing the self-attention mechanism [29], the Attention Model [51] demonstrates the applicability among various VRPs and the efficiency in solving small-scale VRP instances. AM achieves outstanding results in six VRPs including TSP, CVRP, Split Delivery Vehicle Routing (SDVRP), PCTSP, SPCTSP, and OP. Afterward, researchers mainly focus on the two most representative CO problems, TSP and CVRP. Some studies propose more advanced problem-solving strategies [61, 62, 42], others present more efficient network architectures [9, 52], and some focus on designing better loss functions [5, 63] to better escape potential local optima.

By employing a corresponding feasibility mask in each autoregressive construction step, constructive solvers can be effectively applied to most CO problems. Therefore, although many new model structures have been proposed for different CO problems (such as [64] for the Pickup and Delivery Problem (PDP)), ordinary frameworks are usually still feasible. This article adopts the model architectures proposed in POMO [5] and ICAM [16] for most CO problems for convenience. However, for some special CO problems, e.g., MIS, ATSP, and min-max mTSP, conventional constructive solvers are unable to process the problem-specific constraints in their encoders. Therefore, we specifically use one-shot AGNN (consistent with the dividing policy) in sub-MIS, MatNet [41] for sub-ATSP, and DPN [42] to solve sub-min-max mTSP.

### A.3  Heatmap-based Solvers

Recent heatmap-based solvers propose methods to improve the performance of GNN-based frameworks [31, 32]. Attn-MCTS [37] proposes a heuristic method to merge the smaller heatmap obtained from different regions of a TSP instance into a larger heatmap of TSP. More importantly, Attn-MCTS also integrates an effective Monte-Carlo tree search algorithm (MCTS) based on large-scale heatmaps. Therefore, the following heatmap-based works turn to pay more attention to providing better initial heatmaps for the following Monte Carlo tree search [65, 38]. DIMES [35] proposes a meta-learning method based on reinforcement learning for stability in training, while DIFUSCO [39] and T2T [66] use supervised learning to learn a diffusion process [67] of heatmap generation.

### A.4  Neural Divide-and-Conquer Methods

The Table 1 in the main part offers a brief overview of all the existing neural divide-and-conquer methods. These methods and the proposed UDC follow the same two-stage solution generation framework. Instead of being a fully neural method as UDC, some neural divide-and-conquer methods strategically incorporate problem-specific heuristics in either the dividing or conquering stages to

enhance performance or efficiency [22, 23, 21, 13]. Such a setting undermines the possibility to generalize to other CO problems.

LCP [24], H-TSP [20], RBG [23], TAM [12], and GLOP [13] (for CVRP and PCTSP) can use only neural networks in both stages, which enhances the applicability of methods to handle general CO problems without expert experience. However, these methods face challenges in terms of solution qualities. To tackle the instability in unified training [20], all these methods adopt a separate training scheme and we believe there are inherent shortcomings in the separate training schemes. Among them, LCP [24] employs a constructive neural solver (seeder) to generate diversified initial solutions (seeds) and another constructive neural solver (reviser) for sub-path solving. The seeders and revisers are trained separately and the objective functions of its seeders and revisers are not directly related to the objective function of the original CO problem (i.e., belonging to a separate training scheme), so its effectiveness largely depends on randomness.

TAM [12], GLOP [13] (for CVRP and PCTSP), and RBG [23] adopt a similar training scheme, with only using different solvers for the dividing policy (i.e., TAM uses a constructive neural solver, and GLOP chooses a heatmap-based solver). These methods pre-train a conquering policy on a preempted dataset and directly use them as local solvers (i.e., operators) for their initial solution. Such a training scheme fits the Pre-train + Train Dividing variant in Section 5.1 which can hardly converge to a good performance. The pre-trained conquering policy will usually generate sub-optimal sub-solutions so such a training scheme will probably lead to local optima. Moreover, as another drawback of TAM and GLOP, they decompose CVRP and PCTSP into sub-TSP for the pre-trained conquering policy, but such solving framework cannot convert sub-optimal initial partitions into optimal ones, thus affecting the accessibility of optimal solutions.

H-TSP [20] focuses on the TSP problem and designs a lightweight convolutional neural network and clustering mechanism for initial solution generation. Unlike other works, it implements a joint training scheme after pre-trained conquering policies. However, Section 5.1 also demonstrates that the pre-training before a joint (i.e., unified) training scheme will also lead to local optima. Moreover, similar to LCP, it does not align the optimization objectives of the dividing policy with the overall objective function of TSP as well, which further limits its effectiveness.

## B   Detailed Definition

### B.1   Definition of Large-scale CO problems

We refer to the definition in [68] for large-scale CO problems. Large-scale CO problems have more data, decision variables, or constraints. In this article, we have considered more decision variables and data volumes in CO problems like TSP and more constraints in CO problems like MIS (i.e., dense graphs have relatively more constraints). Most large-scale NCO works solve TSPs on scales such as 500, 1000, 2000, etc. [39, 8, 16, 13], so UDC follows this setting in testing TSP and applies it to general CO problems.

### B.2   Definition of Problem & Sub-problem

**TSP**   Traveling salesman problem (TSP) is one of the most representative COPs [69]. Let $\{D = d_{j,k}, j = 1, \ldots, N, k = 1 \ldots, N\}$ be the $N \times N$ cost metrics of a $N$-nodes TSP problem (in practice, transformed from the input node-wise coordinates), where $d_{j,k}$ denotes the cost between nodes $k$ and $j$, the goal is to minimize the following objective function:

$$\text{minimize} \quad f(\boldsymbol{x}) = \sum_{t=1}^{N-1} d_{x_t, x_{t+1}} + d_{x_N, x_1}, \tag{7}$$

where the solution $\boldsymbol{x} = (x_1, x_2, \ldots, x_N)$ $(\tau = N)$ is a permutation of all nodes. All the feasible solutions (i.e., in $\Omega$ of Eq. (1)) satisfy the constraint of node degree being two and containing no loop with lengths less than N.

The sub-TSP in UDC is usually considered as an open-loop TSP problem [70] (i.e., SHPP in GLOP [13]). To make the optimization target of both stages correspond (with objective function $f(\cdot)$ and $f'(\cdot)$, respectively), with the sub-cost metrics $\{D' = d'_{j,k}, j = 1, \ldots, n, k = 1 \ldots, n\}$, the objective function of sub-TSP solution $\boldsymbol{s} = (s_1, s_2, \ldots, s_n)$ is as follow:

$$\text{minimize} \quad f'(\boldsymbol{s}) = \sum_{t=1}^{n-1} d'_{s_t, s_{t+1}} \tag{8}$$

The feasibility set of sub-TSP adds additional selection constraints for the starting and ending points (i.e., the first and last node in the original solution segments). UDC trains the TSP on various scales for every $n$ nodes, from $N$=500 to $N$=1,000 (i.e., for easier implement, we set $n$=100, and sampling $N$ from $\{500, 600, 700, 800, 900, 1,000\}$). Since the nodes in each sub-problem always come from a compact region of the original instance, so we follow the normalization approach in GLOP and conduct a coordinate transformation to coordinates of sub-problems. The specific formula is detailed in Eq. (20).

**CVRP**   CVRP incorporates the vehicle capacity as a restriction while considering the total distance as its objective function. Each CVRP contains a depot and several customers, on a cost matrix $\{D = d_{j,k}, j = 0, \ldots, N, k = 0 \ldots, N\}$, the CVRP can be expressed as follows:

$$\text{minimize} \quad f(\boldsymbol{x}) = \sum_{j=1}^{q} C(\boldsymbol{\rho}^j),$$

$$C(\boldsymbol{\rho}^j) = \sum_{t=0}^{|\boldsymbol{\rho}^j|-1} d_{\rho_t^j, \rho_{t+1}^j} + d_{\rho_{n_j}^j, \rho_0^j} \tag{9}$$

$$\text{subject to} \quad 0 \leq \delta_i \leq C, \quad i = 1, \ldots, n,$$

$$\sum_{i \in \boldsymbol{\rho}^j} \delta_i \leq C, \quad j = 1, \ldots, q,$$

where $\boldsymbol{x}$ is a solution representing the complete route of vehicles and consists of $q$ sub-routes $\boldsymbol{x} = \{\boldsymbol{\rho}^1, \boldsymbol{\rho}^2, \ldots, \boldsymbol{\rho}^q\}$. Each sub-route $\boldsymbol{\rho}^j = (\rho_1^j, \ldots, \rho_{n_j}^j)$, $j \in \{1, \ldots, q\}$ starts from the depot $x_0$ and backs to $x_0$, $n_j$ represents the number of customer nodes in it. $n = \sum_{j=1}^{q} n_j$ is the total

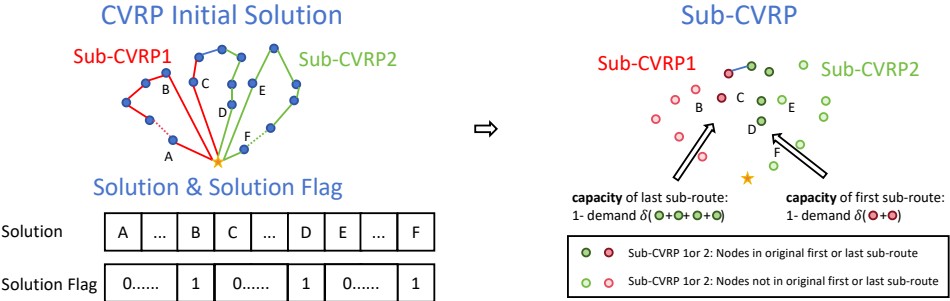

Figure 4: The process of preparing sub-CVRPs.

number of customer nodes; $\delta_i$ denotes the demand of node $i$; $C$ denotes the capacity of the vehicle. In generating CVRP instances in test sets, we follow the common setting in [13, 14], setting $C = 100$ for 500-node CVRP (i.e., CVRP500), $C = 200$ for CVRP1,000, $C = 300$ for CVRP2,000 and larger scale. In training, the capacity $C$ is uniformly sampled from 100 to 200 [6, 16].

Similar to sub-TSP, Sub-CVRP is an open-loop CVRP (not forcing both starting and ending at depot $x_0$ for sub-route $\boldsymbol{\rho}^1$ and $\boldsymbol{\rho}^q$) and also restricts the selection of starting and ending nodes. Moreover, each sub-CVRP instance has a special constraint on the capacity of the first and last sub-routes to ensure the legality of the merged solution. As shown in the right half of Figure 4, the capacity of the first and last sub-route is restricted to be lower than the rest capacity of their connected partial sub-routes. Afterward, each pair of adjunct capacity constraints (shown in Figure 4) is normed to make the sum of 1. We follow the setting of BQ-NCO [14] in representing CVRP solutions, converting the route into a solution and an aligned solution flag which is set to 1 for nodes before a depot.

**OP**  OP assigns each node a prize $\rho_i$ and the goal is to maximize the total prize of all collected nodes, while keeping the total length of the route (a loop through all collected nodes) not surpassing a maximum length $T$. OP contains a depot node serving as the route's starting node. We evaluate the UDC on OP500, OP1,000, and OP2,000 and set the $T$ on all scales to 4. Assume solution route of OP is an $\tau$-length ordered sequence $\boldsymbol{x} = (x_0, \ldots, x_\tau)$ ($x_0$ represents be depot, $\rho_0 = 0$) and $\{D = d_{j,k}, j = 0, \ldots, N, k = 0 \ldots, N\}$ is the $N \times N$ cost metrics of a $N$-nodes TSP problem, the objective function can be formulated as:

$$
\begin{aligned}
\text{maximize} \quad & f(\boldsymbol{x}) = \sum_{i \in \boldsymbol{x}} \rho_i, \\
\text{subject to} \quad & l(\boldsymbol{x}) = \sum_{t=1}^{\tau-1} d_{x_t, x_{t+1}} + d_{x_\tau, x_1} \leq T.
\end{aligned}
\tag{10}
$$

We follow the general OP (distance) (i.e., OP(dist)) setting to generate OP instances [51, 52, 63, 14], whose prize of each node is calculated $\rho_i$ based on its distance to the depot. In practice, we follow the setting in AM [51], $\hat{\rho}_i = 1 + \lfloor 99 \frac{d_{0,i}}{\max_{j=1}^N d_{0j}} \rfloor$, $\rho_i = \frac{\hat{\rho}_i}{100}$. The sub-OP in UDC contains a segment of the original solution while also randomly introducing some unselected nodes. By solving these sub-OPs, the conquering stage gains the potential to improve a low-quality initial solution to optimal. For sub-solution $\boldsymbol{s}$, the objective function of sub-OP is $f'(\boldsymbol{s}) = \sum_{i \in \boldsymbol{s}} \rho_i$. There is no normalization step in the conquering stage when solving OP. Sub-OP restricts the lengths of the generated sub-solution not surpassing the total length of the original sub-OP solution. One of the $\lfloor \frac{N}{n} \rfloor$ sub-OPs contains the depot and the depot is restricted to be included in the sub-solution of this sub-OP.

In testing, solving sub-OP limits the path length not to exceed the original sub-solution length, so continuously executing the conquering stage will result in a gradually shorter total route length $l(\boldsymbol{x}_r)$. So, we count the margin (i.e., $T - l(\boldsymbol{x}_r)$) of the constraint $T$ in the current solution. In each conquering stage, we add the margin $T - l(\boldsymbol{x}_r)$ to the length constraint of a random sub-OP.

**PCTSP**  With a similar optimization requirement compared to OP, PCTSP is also a typical CO problem [51]. PCTSP gives each node a prize $\rho_i$ and a penalty $\beta_i$. The goal is to minimize the total

length of the solution route (a loop through all collected nodes as well) plus the sum of penalties of unvisited nodes. Moreover, as a constraint, PCTSP solutions should collect enough total prizes. The objective function is defined as follows:

$$\text{minimize} \quad f(\boldsymbol{x}) = \sum_{t=1}^{\tau-1} d_{x_t,x_{t+1}} + d_{x_\tau,x_1} + \sum_{i \notin \boldsymbol{x}} \beta_i,$$
$$\text{subject to} \quad \sum_{i \in \boldsymbol{x}} \rho_i \geq 1. \tag{11}$$

We follow the settings in AM [51] and GLOP [13] to generate PCTSP instances, $\rho_i \sim \text{Uniform}(0, \frac{4}{N})$, $\beta_i \sim \text{Uniform}(0, 3 \cdot \frac{K^N}{N})$. We follow the setting in [13], define $K^N = 9, 12, 15$ for $N = 500, 1,000, 2,000$.

Compared to OP, UDC uses a similar way to process sub-PCTSP, introducing random unselected nodes in generated sub-problems and restricting the appearance of the depot. As a similar constraint of sub-OP, sub-PCTSP also restricts the total prize collected in the generated sub-solution to be larger than the total prize in the original sub-solution. Specifically, sub-PCTSP normalizes prizes of each node to $\rho_i'$ by dividing this constraint for better alignment, and the necessity of this procedure is demonstrated in Appendix E with ablation studies. The objective function of sub-PCTSP solution $\boldsymbol{s}$ is as follows:

$$\text{minimize} \quad f'(\boldsymbol{s}) = \sum_{t=1}^{\tau-1} d_{s_t,s_{t+1}} + \sum_{i \notin \boldsymbol{s}} \beta_i,$$
$$\text{subject to} \quad \sum_{i \in \boldsymbol{s}} \rho_i' \geq 1. \tag{12}$$

The existing method GLOP [13] divides the PCTSP into sub-TSP, this problem-specific design not only harms the accessibility to optimal solutions but also impairs the generation ability of the framework to general CO problems.

**SPCTSP** SPCTSP evaluates the capability of the proposed UDC in dealing with uncertainty. As a notable difference to PCTSP, in SPCTSP, the real collected prize of each node only becomes known upon visitation but an expected node prize $\rho_i^* \in \text{Uniform}(0, 2\rho_i)$ is known upfront. In solving SPCTSP with UDC, the expected node prize $\rho_i^*$ is used to process sparse graph $\mathcal{G}_D$, heatmap $\mathcal{H}$, and the encodings of conquering policy, while the real prize $\rho_i$ is only used to process the current states and constraints when constructing solutions of SPCTSP and sub-SPCTSP.

**KP** Besides VRPs, KP is also a typical CO problem which can be formulated as follows:

$$\text{minimize} \quad f(\mathbb{X}) = -\sum_{i \in \mathbb{X}} v_i,$$
$$\text{subject to} \quad \sum_{i \in \mathbb{X}} w_i \leq W, \tag{13}$$

where $\mathbb{X} \subseteq \{1, 2, ..., N\}$ represents the selection item indexes (i.e., KP solution) in a knapsack and $v_i \sim \text{Uniform}(0, 1)$ and $w_i \sim \text{Uniform}(0, 1)$ is the value and weight of candidate objects. In each conquering step, sub-KPs are randomly sampled. Sub-KP and KP are identical in form and objective function, with only distinction on the number of decision variables and total capacity $W$. In training, UDC disables the Reunion stage in the DCR training method for KP, and the capacity $W$ is sampled from 50 to 100. In testing, UDC adopts a similar approach to OP in introducing the margin of maximum capacity (i.e., $W - \sum_{i \in \mathbb{X}} w_i$) to sub-KPs, adding the current margin (i.e., capacity minus current load) to a random sub-KP.

**OVRP** As a variant on CVRP, OVRP calculates the objective function of each sub-route as open-loops. The objective function of OVRP is calculated as follows:

$$\text{minimize} \quad f(\boldsymbol{x}) = \sum_{j=1}^{q} \sum_{t=0}^{|\boldsymbol{\rho}^j|-1} d_{x_t, x_{t+1}}$$

$$\text{subject to} \quad 0 \le \delta_i \le C, \quad i = 1, \dots, n, \tag{14}$$

$$\sum_{i \in \boldsymbol{\rho}^j} \delta_i \le C, \quad j = 1, \dots, q.$$

**ATSP** ATSP is a variant version of TSP with asymmetric cost metrics $\{D = d_{j,k}, j = 0, \dots, N, k = 0 \dots, N\}$ (i.e., ATSP and sub-ATSP have the same objective function as TSP and sub-TSP, respectively). Different from TSP, ATSP instances are inputted as cost metrics directly so this specificity makes it impossible to use conquering models such as POMO [5], ICAM [16], etc. [41] When dealing with sub-ATSP, we employ a matrix-based sub-problem normalization method to normalize the longest item (i.e., edge) in cost metrics of sub-ATSPs to 1. For testing scale, the 1000-node ATSP is already huge in terms of the data size, while the 2000-node data is unable to generate and store in memory, so we refer to the testing scales of GLOP [13], conducting experiments on the 250-node (i.e., ATSP250), the ATSP500, and the ATSP1000.

**MIS** MIS is originally defined on a graph $\mathcal{G} = \{\mathbb{V}, \mathbb{E}\}$ where $N = |\mathbb{V}|$ so in the dividing step we have $\mathcal{G}_D = \mathcal{G}$ directly (without additional operations to construct the sparse graph $\mathcal{G}_D$). For solution set $\mathbb{X} \subseteq \{1, 2, \dots, N\}$, the objective function of MIS is to **maximize** the number of visited nodes in solution (i.e., $f(\mathbb{X}) = |\mathbb{X}|$) while ensuring no edge connections $e \in \mathbb{E}$ between any two visited nodes. In training, follow the MIS instance generation process of ER[700-800] with $p$=0.15 [35]. The nodes in sub-MIS are randomly selected and nodes connecting to visited nodes are constrained as unaccessible. The objective function of a sub-MIS solution $\mathbb{S}$ is $f(\mathbb{S}) = |\mathbb{S}|$. The conquering policy for sub-MIS uses a Heatmap-based solver as well (i.e., AGNN in UDC) with non-autoregressive construction methods.

**Min-max mTSP** As a variant of VRP, the min-max vehicle routing problem (min-max VRP), instead of minimizing the total length of all sub-routes (i.e., min-sum), seeks to reduce the length of the longest one among all the sub-routes (i.e., min-max). Min-max VRP has important application value in real life as well [71, 72] and researchers have also paid attention to solving them with deep learning techniques [62, 42]. The solution of min-max mTSP is defined similarly to CVRP solutions, consisting of **exactly** $M$ **sub-routes** (as a constraint on the number of total sub-routes) $\boldsymbol{x} = \{\boldsymbol{\rho}^1, \boldsymbol{\rho}^2, \dots, \boldsymbol{\rho}^M\}$. The objective function of min-max mTSP is to minimize the length of the longest sub-route, which is as follows:

$$\text{minimize} \quad f(\boldsymbol{x}) = \max_{i=1}^{M} C(\boldsymbol{\rho}^j),$$

$$C(\boldsymbol{\rho}^j) = \sum_{t=0}^{|\boldsymbol{\rho}^j|-1} d_{\rho_t^j, \rho_{t+1}^j} + d_{\rho_{n_j}^j, \rho_0^j} \tag{15}$$

Similar to the normal min-sum objective function, the min-max objective function can also be decomposed to independent objective functions of sub-problems so UDC can be evaluated on it. However, as the difference, the optimization of sub-min-max mTSP should encounter the lengths of sub-routes in adjunct sub-problems. For a sub-min-max mTSP with lengths of sub-routes in adjunct sub-problems being $l_b$ and $l_e$, the objective function of sub-min-max mTSP solution with $m$ sub-routes $\boldsymbol{s} = \{\boldsymbol{\rho}'^{,1}, \boldsymbol{\rho}'^{,2}, \dots, \boldsymbol{\rho}'^{,m}\}$ is calculated after adding $l_b$ to the first sub-route length $C(\boldsymbol{\rho}^1)$ and $l_e$ to $C(\boldsymbol{\rho}^m)$ as follows:

$$\text{minimize} \quad f(\boldsymbol{s}) = \max_{i=1}^{m} \{C(\boldsymbol{\rho}'^{,1}) + l_b, C(\boldsymbol{\rho}'^{,2}), \dots, C(\boldsymbol{\rho}'^{,m}) + l_e\}, \tag{16}$$

Note that only holding two adjunct sub-problems unchanged can make the optimization of the sub-min-max mTSP consistent with the original min-max mTSP. So, to maintain optimality, the testing stage will not update adjacent sub-min-max-mTSP simultaneously.

# C  Method Details

## C.1  Dividng Stgae: Graph Construction

Light-weight GNNs require constructing a graph as input (the sparse graph $\mathcal{G}_D$). Some CO problems involved in this paper naturally have graph-like input (e.g., the MIS problem) so their sparse graph $\mathcal{G}_D = \mathcal{G}$. However, the inputs of most other CO problems are coordinates or matrics. So, for the VRPs, we adopt the graph construction method in heatmap-based solvers [31] and GLOP [13], generally using the KNN method to connect edges in the edge set $\mathbb{E} \in \mathcal{G}_d$. For CVRP, OVRP, ATSP, OP, PCTSP, SPCTSP, and min-max mTSP (min-max mTSP specifically uses dummy depots for constraints [62]), we directly use a similar method to link edges based on KNN. In addition, for the KP without neighborhood properties, we use $1 - \frac{d'_{i,j}}{\max_{q \in 1, \dots, N} d'_{i,q}}$ as the weight of $(i,j)$ for neighborhood selection where $d'_{i,j} = \frac{v_i + v_j}{w_i + w_j}$ and use the KNN for edge connection as well. In the construction of all the involved CO problems (excluding MIS), we set neighborhood size $K = 100$, so the number of edges in the graph is limited to $|\mathbb{E}| = \mathcal{O}(KN)$.

## C.2  Dividing Policy: Anisotropic Graph Neural Networks

In modeling the dividing policy, we follow previous heatmap-based methods [39] on the choice of neural architectures. The graph neural network of UDC is an anisotropic GNN with an edge gating mechanism [40]. Let $\boldsymbol{h}_i^\ell$ and $\boldsymbol{e}_{i,j}^\ell$ denote the $d$-dimensional node and edge features at layer $\ell$ associated with node $i$ and edge $(i,j)$, respectively. The features at the next layer are propagated with an anisotropic message-passing scheme:

$$\boldsymbol{h}_i^{\ell+1} = \boldsymbol{h}_i^\ell + \alpha(\text{BN}(\boldsymbol{U}^\ell \boldsymbol{h}_i^\ell + \mathcal{A}_{j \in \mathcal{N}_i}(\sigma(\boldsymbol{e}_{i,j}^\ell) \odot \boldsymbol{V}^\ell \boldsymbol{h}_j^\ell))), \tag{17}$$

$$\boldsymbol{e}_{i,j}^{\ell+1} = \boldsymbol{e}_{i,j}^\ell + \alpha(\text{BN}(\boldsymbol{P}^\ell \boldsymbol{e}_{i,j}^\ell + \boldsymbol{Q}^\ell \boldsymbol{h}_i^\ell + \boldsymbol{R}^\ell \boldsymbol{h}_j^\ell)). \tag{18}$$

where $\boldsymbol{U}^\ell, \boldsymbol{V}^\ell, \boldsymbol{P}^\ell, \boldsymbol{Q}^\ell, \boldsymbol{R}^\ell \in \mathbb{R}^{d \times d}$ are the learnable parameters of layer $\ell$, $\alpha$ denotes the activation function (we use SiLU [73] in this paper), BN denotes the Batch Normalization operator [74], $\mathcal{A}$ denotes the aggregation function (we use mean pooling in this paper), $\sigma$ is the sigmoid function, $\odot$ is the Hadamard product, and $\mathcal{N}_i$ denotes the outlines (neighborhood) of node $i$. We use a 12-layer ($L$=12) AGNN with a width of $d = 64$. The total space and time complexity of AGNN is $\mathcal{O}(KNd)$. In training all involved CO problems, the learning rate of the dividing policy is set to $lr = 1e - 4$.

**$T$-revisit initial solution decoding.** Next, based on the Eq. (4), the final layer edge representation $\boldsymbol{e}_{i,j}^L$ of AGNN is used to generate the heatmap. Heatmap-based solvers generally use an MLP to process the $\boldsymbol{e}_{i,j}$. To integrate partial solution-related representations in the heatmap, UDC has designed an additional $T$-revisit method, which will re-generate the heatmap by an additional $T-1$ times using MLPs based on the existing partial solutions after multiple heatmap decoding steps. This operation increases the time and space complexity to $\mathcal{O}(TKNd)$, but significantly improves the solution quality. Enabling the $T$-revisit method, the total initial solution generation strategy is shown below.

$$\pi_d(\boldsymbol{x}_0 | \mathcal{G}, \Omega, \phi) = \begin{cases} p(x_{0,1}) \prod_{c=0}^{T-1} \left[ p(\mathcal{H} | \mathcal{G}, \Omega, \phi, x_{0,1:\lfloor \frac{N}{T} \rfloor c+1}) \prod_{t=2}^{\tau} \frac{\exp(\mathcal{H}_{x_{0,t-1},x_{0,t}})}{\sum_{i=t}^N \exp(\mathcal{H}_{x_{0,t-1},x_{0,i}})} \right], & \text{if } \boldsymbol{x}_0 \in \Omega \\ 0, & \text{otherwise} \end{cases}. \tag{19}$$

It represents decoding solutions from the heatmap $\mathcal{H} \sim p(\mathcal{H} | \mathcal{G}, \Omega, \phi, x_{0,1:\lfloor \frac{N}{T} \rfloor c+1})$ and a new heatmap will generates every $\lfloor \frac{N}{T} \rfloor$ steps. For CO problems without a depot (i.e., TSP, KP, MIS, ATSP), $p(x_{0,1}) = \frac{1}{N}$ and for other CO problems, $x_{0,1} =$ depot and $p(x_{0,1}) = 1$. In training UDC, we set $T = \lfloor \frac{N}{n} \rfloor$. In testing, we generally set $T = \lfloor \frac{N}{n} \rfloor$ as well for 500-node to 4,999-node random instances. For efficiency, we set $T = 10$ for very large-scale CO instances and CVRPLib-XXL datasets.

## C.3 Conquering Stage: Sub-problem Preparation

The detail of decomposing CO problems to their corresponding sub-CO problems is shown in B. Generally, to generate the constraints of different sub-CO problems at this step, we need to **1)** Keep the already determined solutions and likes unchanged (e.g., starting point in VRP, points connected to visited nodes in MIS) **2)** Dispatch the original constraints to the current sub-problems (capacity of CVRP, capacity of KP, maximum length of OP, etc.). Usually, this kind of constraint is calculated from nodes in the original sub-solution. **3)** In testing, consider the margin of constraints in the original CO problems (i.e., KP, OP, PCTSP, SPCTSP) to ensure the accessibility of optimal solutions.

## C.4 Conquering Stage: Normalization

As presented in B, TSP, ATSP, CVRP, OVRP, and min-max mTSP conduct normalization on the inputs of their sub-problems. ATSP normalizes the distance matrix by dividing by the maximum in each sub-ATSP cost matrix (i.e., $d'_{i,j} = \frac{d_{i,j}}{max_{o,q \in \{1,...,N\}} d_{o,q}}$) and other sub-CO problems are normalized following the method in GLOP [13]. Let $(x'_i, y'_i)$ denotes the coordinates of the $i$-th node after normalization, $x_{\max}$, $x_{\min}$, $y_{\max}$, and $y_{\min}$ denote the bounds of an sub-VRP instance $\{(x_1, y_1), \ldots, (x_n, y_n)\}$ (i.e., $x_{\max} = \max_{i \leq n} x_i$, et al.). Then, the coordinate transformation of coordinate $(x_i, y_i)$ is formulated as follows:

$$
\begin{aligned}
x'_i &= \begin{cases} sc(x_i - x_{\min}) & \text{if } x_{\max} - x_{\min} > y_{\max} - y_{\min}, \\ sc(y_i - y_{\min}) & \text{otherwise}, \end{cases} \\
y'_i &= \begin{cases} sc(y_i - y_{\min}) & \text{if } x_{\max} - x_{\min} > y_{\max} - y_{\min}, \\ sc(x_i - x_{\min}) & \text{otherwise}, \end{cases}
\end{aligned}
\tag{20}
$$

where the scale coefficient $sc = \frac{1}{\max(x_{\max} - x_{\min}, y_{\max} - y_{\min})}$. Besides normalizing the coordinates, sub-SPCTSP and sub-PCTSP normalize prizes of nodes by dividing the total collected prize of the current sub-solution.

## C.5 Conquering Stage: Conquering Policy

UDC adopts 5 solvers as the conquering policy, including AGNN [35] for MIS, ICAM [16] for TSP, CVRP, OVRP, and KP, POMO [5] for PCTSP, SPCTSP, OP, MatNet [41] for ATSP, and DPN [42] for min-max mTSP. We employ 12-layer and 256-dimensional AGNN, 6-layer ICAM without AAFT, 6-layer POMO, and 5-layer MatNet, for these experiments respectively. In training, the learning rate of all these solvers is set to $lr = 1e - 4$ as well.

The baseline calculation method in training conquering policies generally follows POMO [5] (sampling $\beta$ sub-solutions for each sub-CO instance), but the generation process of the $\beta$ sub-solutions is different for each CO problem. Some sub-CO problems (e.g., sub-TSP, sub-CVRP, sub-OVRP, sub-OP) have symmetry (i.e., flipping sub-solutions will not change their objective values) so we enable a **two sides conquering baseline**, sampling $\frac{\beta}{2}$ solutions on both the starting and ending nodes. For other CO problems, we sample $\beta$ solutions from the starting point (e.g., the first randomly selected vertex in sub-MIS and the starting node of sub-ATSP). Please refer to the second to last line in Table 7 for whether the conquering policy enables the two sides conquering baseline in solving each CO problem. In testing, the conquering policy in the UDC samples only one sub-solution (i.e., $\beta = 1$) for asymmetric sub-CO problems and sampling only two sub-solutions (i.e., $\beta = 2$) for symmetric ones.

**Reason for using different solvers for different CO problems** Using the same model for conquering policy is an ideal choice. However, the experiment of UDC involves 10 CO problems, and some CO problems, like min-max mTSP [42] and ATSP [41], require specific model structures to process their inputs and constraints. Consequently, there is no constructive solver that can be used for all 10 involved CO problems. For each CO problem, UDC adopts the best-performance constructive solver for the conquering policy (i.e., summarized in Table 7).

**UDC exhibits flexibility in choosing a constructive solver for conquering policy.** For well-developed CO problems like TSP and KP, there are multiple constructive solvers available. Section 5.2 and Appendix E.6 present an ablation study on the selection of these constructive solvers for TSP

and KP, respectively. Both sets of results indicate that the final performance of UDC is not sensitive to the model selection for the conquering policy. Therefore, when generalizing UDC to new CO problems, it may not be necessary to specifically choose which available constructive solver to model the conquering policy.

## C.6 Total Algorithm

The total algorithm of training UDC is presented in Algorithm 1. The total algorithm of testing UDC on instances is presented in Algorithm 2.

---

**Algorithm 1** UDC: Training dividing and conquering policy on general large-scale CO problems.

---

1: **Input:** A CO instance $\mathcal{G}$, number of node $N$; scale of sub-problems $n$; the number of sampled initial solution $\alpha$; the number of sampled sub-solution $\beta$; dividing policy $\phi$, conquering policy $\theta$
2: **Output:** parameter $\phi, \theta$
3: Generate sparse graph $\mathcal{G}_D$
4: Generate initial Heatmap $\mathcal{H}$ based on $p(\mathcal{H}|\mathcal{G}_D, \Omega, \phi)$
5: Sample $\alpha$ initial tours: $\{\boldsymbol{x}_0^1, \dots, \boldsymbol{x}_0^\alpha\}$ based on $\pi_c$ in Eq. (4). (Eq. (19) for $T$-revisit enabled.)
6: **for** $i \in \{1, 2\}$ **do**
7:     Initialize the decomposition point: $p \leftarrow \frac{n}{2} * i$
8:     Calculate the number of sub-tours decomposed from a tour: $C = \lfloor \frac{N}{n} \rfloor$
9:     $\{\mathcal{G}_1^{i-1}, \dots, \mathcal{G}_C^{i-1}, \dots, \mathcal{G}_{\alpha C}^{i-1}\} \leftarrow$
10:         $\{\{\boldsymbol{x}_{0,p:p+n}^1, \dots, \boldsymbol{\pi}_{0,p+n(C-1):p+nC}^1\}, \dots, \{\boldsymbol{x}_{0,p:p+n}^\alpha, \dots, \boldsymbol{x}_{0,p+n(C-1):p+nC}^\alpha\}\}$
11:     Apply coordinate transformation and constraint normalization to $\{\mathcal{G}_1^{i-1}, \dots, \mathcal{G}_{\alpha C}^{i-1}\}$
12:     **for** $k \in \{1, \dots, \alpha C\}, j \in \{1, \dots, \beta\}$ **do**
13:         Sample $\boldsymbol{s}_k^{i,j} \sim \pi_c(\cdot|\mathcal{G}_k^{i-1}, \Omega_k^{i-1}, \theta)$
14:         **if** i = 1 **then**
15:             Update conquering policy $\theta \leftarrow \theta + \nabla_\theta \mathcal{L}_{c1}(\mathcal{G})$
16:         **end if**
17:         **if** i = 2 **then**
18:             Update conquering policy $\theta \leftarrow \theta + \nabla_\theta \mathcal{L}_{c2}(\mathcal{G})$
19:         **end if**
20:     **end for**
21:     **for** $k \in \{1, \dots, \alpha C\}$ **do**
22:         Select best sub-solution among $\beta$ samples: $\hat{\boldsymbol{s}}_k^i = \text{argmax}_{\boldsymbol{s}' \in \{\boldsymbol{s}_k^{i,1}, \dots, \boldsymbol{s}_k^{i,\beta}\}} f(\boldsymbol{s}')$
23:         Substitute the $\boldsymbol{s}_k^{i,j}$ with no improvement on $f(\hat{\boldsymbol{s}}_k^i)$ in sub-CO problems to the original solution.
24:         Merge solutions: $\{\boldsymbol{x}_i^1, \dots \boldsymbol{x}_i^\alpha\} = \{\text{Concat}(\hat{\boldsymbol{s}}_1^i, \dots, \hat{\boldsymbol{s}}_C^i), \dots, \text{Concat}(\hat{\boldsymbol{s}}_{(\alpha-1)C+1}^i, \dots, \hat{\boldsymbol{s}}_{\alpha C}^i)\}$
25:     **end for**
26: **end for**
27: Update dividing policy $\phi \leftarrow \phi + \nabla_\phi \mathcal{L}_d(\mathcal{G})$

---

## C.7 MDP of the dividing and conquering stages

In addition to the algorithms in Appendix C.6, this section will provide necessary descriptions of the training objective of both policies and the implicit MDP for the two stages.

It should be noted that the training objectives and MDP are not designed originally from UDC, they are also adopted in a series of neural divide-and-conquer methods [13, 20].

**Objective functions of optimizing the two networks** . For a CO problem (or a sub-CO problem) with the objective function $f(\cdot)$, the goal of training both networks is to maximize the reward functions of their corresponding MDP. The reward is $-f(\boldsymbol{x}_2)$ for the dividing policy (AGNN in UDC), $-f(\boldsymbol{s}^1)$ for the constructive solver (conquering policy) in the Conquer step (i.e., the first conquering stages), and $-f(\boldsymbol{s}^2)$ in the Reunion step.

**Algorithm 2** UDC: Solving general large-scale CO problems.

---

1: **Input:** A CO instance $\mathcal{G}$, number of node $N$; scale of sub-problems $n$; the number of sampled solution $\alpha$; number of conquering stage $r$
2: **Output:** The best solution $\hat{\boldsymbol{x}}$
3: Generate sparse graph $\mathcal{G}_D$
4: Generate initial Heatmap $\mathcal{H}$ based on $p(\mathcal{H}|\mathcal{G}_D, \Omega, \phi)$
5: Sample $\alpha$ initial tours: $\{\boldsymbol{x}_0^1, \ldots, \boldsymbol{x}_0^\alpha\}$ based on $\pi_c$ in Eq. (4). (Eq. (19) for $T$-revisit enabled.) $\beta \leftarrow 2$ for TSP, ATSP, OP, PCTSP, SPCTSP.
6: **for** $i \in \{1, \ldots, r\}$ **do**
7:    **if** $i \leq 2$ **then**
8:       Initialize the decomposition point: $p \leftarrow \frac{n}{2} * i$
9:    **end if**
10:    **if** $i \geq 3$ **then**
11:       Initialize the decomposition point: $p \sim \text{Uniform}(1, n)$
12:    **end if**
13:    Calculate the number of sub-tours decomposed from a tour: $C = \lfloor \frac{N}{n} \rfloor$
14:    $\{\mathcal{G}_1^{i-1}, \ldots, \mathcal{G}_C^{i-1}, \ldots, \mathcal{G}_{\alpha C}^{i-1}\} \leftarrow$
15:       $\{\{\boldsymbol{x}_{i-1,p:p+n}^1, \ldots, \boldsymbol{x}_{i-1,p+n(C-1):p+nC}^1\}, \ldots, \{\boldsymbol{x}_{i-1,p:p+n}^\alpha, \ldots, \boldsymbol{x}_{i-1,p+n(C-1):p+nC}^\alpha\}\}$

16:    Apply coordinate transformation and constraint normalization to $\{\mathcal{G}_1^{i-1}, \ldots, \mathcal{G}_{\alpha C}^{i-1}\}$
17:    **for** $k \in \{1, \ldots, \alpha C\}, j \in \{1, \ldots, \beta\}$ ($\beta$=1 or 2, see Appendix C.5) **do**
18:       Sample $\boldsymbol{s}_k^{i,j} \sim \pi_c(\cdot|\mathcal{G}_k^{i-1}, \Omega_k^{i-1}, \theta)$
19:    **end for**
20:    **for** $k \in \{1, \ldots, \alpha C\}$ **do**
21:       $\hat{\boldsymbol{s}}_k^i = \text{argmax}_{\boldsymbol{s}' \in \{\boldsymbol{s}_k^{i,1}, \ldots, \boldsymbol{s}_k^{i,\beta}\}} f(\boldsymbol{s}')$
22:       Substitute the $\boldsymbol{s}_k^{i,j}$ with no improvement on $f(\hat{\boldsymbol{s}}_k^i)$ in sub-CO problems to the original solution.
23:       Merge solutions: $\{\boldsymbol{x}_i^1, \ldots \boldsymbol{x}_i^\alpha\} \leftarrow \{\text{Concat}(\hat{\boldsymbol{s}}_1^i, \ldots, \hat{\boldsymbol{s}}_C^i, \boldsymbol{x}_{i-1,p+nC+1,p}^1)$
24:              $, \ldots, \text{Concat}(\hat{\boldsymbol{s}}_{(\alpha-1)C+1}^i, \ldots, \hat{\boldsymbol{s}}_{\alpha C}^i, \boldsymbol{x}_{i-1,p+nC+1,p}^\alpha)\}$
25:    **end for**
26:    $\hat{\boldsymbol{x}} = \text{argmax}_{\boldsymbol{x} \in \{\boldsymbol{x}_i^1, \ldots, \boldsymbol{x}_i^\alpha\}} f(\boldsymbol{x})$
27: **end for**

---

**MDP for the dividing stage:** The MDP $\mathcal{M}_d = \{\mathcal{S}_d, \mathcal{A}_d, \boldsymbol{r}_d, \mathcal{P}_d\}$ of the dividing stage can be represented as follows:

*State.* State $st_d \in \mathcal{S}_d$ represents the current partial solution. The state in the $t$-th time step is the current partial solution with $t$ nodes $st_{d,t} = \boldsymbol{x}_{0,t} = (x_1, x_2, ..., x_t)$. $st_{d,0}$ is empty and $st_{d,T} = \boldsymbol{x}_0$.

*Action & Transaction.* The action is to select a node at time step $t$, i.e., $a_{d,t} = x_{t+1}$. The chosen node needs to ensure that the partial solution $st_{d,t+1} = (x_1, x_2, ..., x_t, x_{t+1})$ is valid (i.e., $st_{d,T} \in \Omega$).

*Reward.* Every single time step has the same reward $r_{d,t} = -f(\boldsymbol{x}_2)$ which is the objective function value of the final solution $\boldsymbol{x}_2$ after the whole DCR process.

*Policy.* The policy $\pi_d$ is shown in Eq. (4) (Eq. (19) for $T$-revisit enable).

**MDP for the conquering stage:** The MDP $\mathcal{M}_c = \{\mathcal{S}_c, \mathcal{A}_c, \boldsymbol{r}_c, \mathcal{P}_c\}$ of any single conquering stage is represented similarly as follows:

*State.* Each state $st_c \in \mathcal{S}_c$ represents the current partial solution. The state in the $t$-th time step is the current partial solution with $t$ nodes $st_{c,t} = \boldsymbol{s}_t = (s_1, s_2, ..., s_t)$. $st_{c,0}$ is empty and $st_{c,T} = \boldsymbol{s}$.

*Action & Transaction.* The action is to select a node at time step $t$ as well, i.e., $a_{c,t} = s_{t+1}$.

*Reward.* The reward in each time step becomes the objective value of sub-CO solution $\boldsymbol{s}$, i.e., $r_{c,t} = -f(\boldsymbol{s})$.

*Policy.* The policy $\pi_c$ is shown in Eq. (5).

## C.8 Time & Space Complexity

Table 6: Comparison on time and space complexity.

|  | LEHD | ICAM-single-start | ICAM-N-start | UDC-$\boldsymbol{x}_2$ | UDC-$\boldsymbol{x}_r$ |
|---|---|---|---|---|---|
| Time | $\mathcal{O}(N^3)$ | $\mathcal{O}(N^2)$ | $\mathcal{O}(N^3)$ | $\mathcal{O}(\alpha TKN + n^2)$ | $\mathcal{O}(\alpha TKN + rn^2)$ |
| Space | $\mathcal{O}(N^2)$ | $\mathcal{O}(N^2)$ | $\mathcal{O}(N^2)$ | $\mathcal{O}(\alpha KN + \alpha\lfloor\frac{N}{n}\rfloor n^2)$ | $\mathcal{O}(\alpha KN + \alpha\lfloor\frac{N}{n}\rfloor n^2)$ |

With a lightweight GNN for global partition and constructive solvers for parallel small-scale sub-problems, both the time and space complexities of UDC are reduced compared to single-stage SL-based sub-path solvers and single-stage RL-based constructive-based solvers. When sampling $\alpha$ $T$-revisit enabled initial solutions ($T$=1 for disabling the $T$-revisit), the dividing stage of UDC has a time complexity of $\mathcal{O}(\alpha TKN)$ and a space complexity of $\mathcal{O}(\alpha KN)$.

Both the time and space complexity of a single conquering stage on a single sub-problem is $\mathcal{O}(n^2)$. There are $\alpha\lfloor\frac{N}{n}\rfloor$ parallel sub-problems for each instance so the complexity should be $\alpha\lfloor\frac{N}{n}\rfloor\mathcal{O}(n^2)$. However, the $\alpha\lfloor\frac{N}{n}\rfloor$ sub-problems can be processed in parallel (as batch size = $\alpha\lfloor\frac{N}{n}\rfloor$) in GPUs, so the total time complexity of UDC with $r$ conquering stages can be considered as $\mathcal{O}(\alpha TKN + rn^2)$ and the space complexity is $\mathcal{O}(\alpha KN + \alpha\lfloor\frac{N}{n}\rfloor n^2)$.

Sub-path-based constructive solver adopts a decoder-only model structure so for a $\tau$-length solution, the time complexity is $\mathcal{O}(\tau N^2 d) = \mathcal{O}(N^3)$ ($d$ is the dimension of embeddings, we regard $\tau = \mathcal{O}(n)$) and the space complexity is $\mathcal{O}(N^2)$. ICAM is a representative RL-based constructive solver, adopting an encoder-decoder-based model structure. The space complexity is $\mathcal{O}(N^2)$ and the time complexity is $\mathcal{O}(\tau N d) = \mathcal{O}(N^2)$ for single-start ICAM and $\mathcal{O}(\tau N^2 d) = \mathcal{O}(N^3)$ for $N$-start (multi-start) ICAM. The comparison of time and space complexity is listed in Table 6, UDC generally adopts the parameter of $T = 10$, $\alpha = 1$, $K = 100$ in testing. So on CO problems with $N \geq 1,000$, the proposed UDC-$\boldsymbol{x}_2$ demonstrates less inference complexity compared to $\mathcal{O}(N^2)$ and exhibits superior efficiency. Empirical results especially very-large-scale experiments ($N \geq 5,000$) in Table 11 and Appendix D.5 prove the above statements.

# D Experiment Details

## D.1 Hyperparameter

Hyperparameters of all 10 involved CO problems are listed in Table 7. Training UDC on most CO problems adopts the varying-size [45] and varying-constraint [6] setting. ICAM and POMO used for conquering policy are 6-layers. DPN is 3-layers, and MatNet is 5-layer. AGNN for conquering sub-MIS is 12-layer with $d = 256$. In training UDC for TSP, we don't conduct the varying scale training in the first 50 epochs with $\alpha = 80$.

Table 7: Hyper parameter of training CO problems.

| Hyperparameter | | | | |
|---|---|---|---|---|
| | TSP | PCTSP,SPCTSP | CVRP,OVRP | OP |
| Varying training size $N$ | 500-1,000 | 500-1,000 | 500-1,000 | 500-1,000 |
| Varying training constraint | - | 9-12 | 50-100 | - |
| Sub-path scale $n$ | 100 | 100 | 100 | 100 |
| $\alpha$ in training | 40 | 30 | 40 | 40 |
| $\beta$ in training | 50 | 50 | 40 | 50 |
| Epoch size | 1,000 | 1,000 | 1,000 | 1,000 |
| Epoch-time | 28m | 29m | 42m | 22m |
| Total epochs | 500 | 500 | 200 | 500 |
| DCR enable | TRUE | TRUE | TRUE | TRUE |
| Two sides conquering baseline | TRUE | TRUE | FALSE | TRUE |
| Conquering Policy | ICAM | POMO | ICAM | POMO |
| Hyperparameter | | | | |
| | KP | MIS(ER[700-800]) | min-max mTSP | ATSP |
| Varying training size $N$ | 500-1,000 | 700-800 | 500-1,000 | 250-500 |
| Varying training constraint | 50-100 | - | 30-50 | - |
| Sub-path scale $n$ | 100 | 200 | 100 | 50 |
| $\alpha$ in training | 50 | 35 | 25 | 30 |
| $\beta$ in training | 100 | 35 | 30 | 50 |
| Epoch size | 1,000 | 1,000 | 1,000 | 1,000 |
| Epoch-time | 25m | 7m | 42m | 29m |
| Total epochs | 120 | 1,000 | 200 | 300 |
| DCR enable | FALSE | TRUE | TRUE | TRUE |
| Two sides conquering baseline | FALSE | FALSE | FALSE | FALSE |
| Conquering Policy | ICAM | AGNN | DPN | MatNet |

It can be seen that UDC adopts similar setups to solve different problems. The third to last and the second to last rows of Table 7 are related to the environment and the last row is unable to use a consistent setup and some flexibility in choices. Ablation studies in Section 5.2 and Appendix E discuss the setting of some listed hyperparameters, such as $\alpha$ in testing, DCR enable=False for KP, and the selection of conquering Policy in KP and TSP.

## D.2 Experiment on OVRP, MIS, ATSP, KP, and min-max mTSP

The main part of this paper has evaluated the proposed UDC on large-scale instances of 5 representative CO problems. In this section. We will further evaluate the proposed UDC on more CO problems. OVRP is a variant of CVRP that has been widely involved in recent multi-task NCO work [75, 76]. We introduce this problem to test UDC's ability to handle open loops and distant point-to-point relationships based on CVRP. MIS is a classic CO problem widely used to evaluate heatmap-based solvers [35], which can test the capability of UDC to handle CO problems on graphs. The experiment on ATSP can evaluate the flexibility of the UDC framework and its ability to process matrix-based data. We also employ KP to test the ability of UDC to handle CO problems without neighborhood properties. Most importantly, min-max mTSP is introduced to test the ability of UDC to handle problems with different aggregation functions (i.e., the max aggregation function) except for min-sum (i.e., objective functions of all the else 9 CO problems). For OVRP, KP, ATSP, and min-max mTSP in this section, UDC works to minimize rather than maximize (like OP and MIS) their objective functions.

**Experiment on OVRP:** The experiment result is shown in Table 8. Compared to the current best NCO method [75] POMO (Trained on OVRP100 with settings in [75]), UDC demonstrates significant superiority in large-scale OVRP problems.

Table 8: Objective function (Obj.), Gap to the best algorithm (Gap), and solving time (Time) on 500-node (128 instances), 1,000-node (128 instances), and 2,000-node OVRP (16 instances).

| | OVRP500 | | | OVRP1,000 | | | OVRP2,000 | | |
|---|---|---|---|---|---|---|---|---|---|
| Method | Obj.↓ | Gap | Time | Obj.↓ | Gap | Time | Obj.↓ | Gap | Time |
| LKH3 | 23.51 | - | 16m | 28.96 | - | 32m | 39.88 | - | 27m |
| POMO | 28.73 | 22.21% | 1.5m | 59.26 | 104.61% | 16m | 108.82 | 172.90% | 16m |
| UDC-$x_2$($\alpha$=50) | 25.82 | 9.86% | 1.4m | 33.01 | 13.97% | 3m | 51.11 | 28.17% | 42s |
| UDC-$x_{50}$($\alpha$=50) | 24.39 | 3.77% | 7.9m | 29.95 | 3.41% | 15.5m | 44.19 | 10.82% | 4.5m |
| UDC-$x_{250}$($\alpha$=50) | 24.18 | 2.85% | 34.5m | 29.66 | 2.39% | 1.1h | 43.35 | 8.71% | 20m |

**Experiment on MIS:** The MIS problem is a widely adopted CO problem for heatmap-based solvers. UDC can also solve the MIS problem by employing AGNN as the conquering policy. Following DIFUSCO, we evaluate the proposed UDC on the Erdős-Rényi (ER) [77] dataset. We evaluate the MIS on ER[700-800] with $p$=0.15. We use the test set provided in [35], and the results on it are shown in Table9. UDC can also achieve good results on MIS after several conquering stages. Compared to the heatmap-based solver that originally relied on search algorithms, UDC shows better efficiency.

Table 9: Performance on 128-instance ER[700-800] test set (provided in DIMES [35]). All the results are reported in DIMES [35] DIFUSCO [39] and T2T [66]. For MIS, the larger objective value (Obj.) is better.

| | ER-[700-800] | | |
|---|---|---|---|
| Method | Obj.↑ | Gap. | Time |
| KaMIS* | 44.87 | - | 52.13m |
| Gurobi* | 41.38 | 7.78% | 50.00m |
| hline Intel* | 38.8 | 13.53% | 20.00m |
| DGL* | 37.26 | 16.96% | 22.71m |
| LwD* | 41.17 | 8.25% | 6.33m |
| DIFUSCO-greedy* | 38.83 | 13.46% | 8.80m |
| DIMES-greedy* | 38.24 | 14.78% | 6.12m |
| T2T-greedy* | 39.56 | 11.83% | 8.53m |
| UDC-$x_2$($\alpha$=50) | 41.00 | 8.62% | 40s |
| DIMES-MCTS* | 42.06 | 6.26% | 12.01m |
| DIFUSCO-MCTS* | 41.12 | 8.36% | 26.67m |
| T2T-MCTS* | 41.37 | 7.80% | 29.73m |
| UDC-$x_{50}$($\alpha$=50) | 42.88 | 4.44% | 21.05m |

**Experiment on ATSP:** On ATSP, we compare the proposed UDC to traditional solver OR-Tools, constructive solver MatNet, and neural divide-and-conquer method GLOP. Results on ATSP250 to ATSP1,000 are shown in Table 10 and results of MatNet and GLOP are reported from GLOP [13]. We follow the data generation method in MatNet [41] and such a data generation method is unavailable on larger scales (i.e., ATSP2,000). Results demonstrate the capability of UDC on large-scale ATSP and the superiority of UDC compared to GLOP. The advantage of performance contributes to the unified training scheme (ATSP of GLOP can be regarded as the separate training UDC.).

Table 10: Objective function (Obj.), Gap to the best algorithm (Gap), and solving time (Time) on 250-node (128 instances), 500-node (128 instances), and 1,000-node ATSP (16 instances). $\alpha = 20$ for ATSP250 and ATSP500 and $\alpha = 10$ for ATSP1,000.

| Methods | ATSP250 | | | ATSP500 | | | ATSP1,000 | | |
|---|---|---|---|---|---|---|---|---|---|
| | Obj.↓ | Gap | Time | Obj.↓ | Gap | Time | Obj.↓ | Gap | Time |
| OR-Tools | 1.94 | 5.32% | 1m | 2.00 | 1.57% | 2m | 2.08 | 0.14% | 2m |
| MatNet* | 4.49 | 143.89% | - | - | - | - | - | - | - |
| GLOP* | 2.10 | 14.07% | - | - | - | - | 2.79 | 34.39% | - |
| UDC-$x_2$ | 1.96 | 6.63% | 21s | 2.05 | 3.92% | 39s | 2.16 | 3.85% | 6s |
| UDC-$x_{50}$ | 1.84 | - | 7m | 1.97 | - | 13m | 2.08 | - | 1.5m |

**Experiment on KP:** On KP, we involve OR-Tools (generally optimal on KP), POMO, and BQ as baselines. As shown in Table 11, compared to BQ and POMO, both the UDC-$x_{50}(\alpha=1)$ and UDC-$x_{250}(\alpha=1)$ exhibit the best learning-based performance on KP500 and KP1,000. Moreover, UDC demonstrates outstanding time efficiency on KP1,000 and larger-scale KPs.

Table 11: Objective function (Obj.), Gap to the best algorithm (Gap) on 500-node, 1,000-node (128 instances), and 2,000-node, 5,000-node KP (16 instances). OR-Tools can generate solutions in real time. The best learning-based method is marked by shade.

| Method | KP500,$W$=50 | | | KP1,000,$W$=100 | | |
|---|---|---|---|---|---|---|
| | Obj.↓ | Gap | Time | Obj.↓ | Gap | Time |
| OR-Tools | 128.3690 | - | - | 258.2510 | - | - |
| BQ | 128.2786 | 0.07% | 37s | 258.0553 | 0.08% | 4.4m |
| POMO | 128.3156 | 0.04% | 5s | 254.5366 | 1.44% | 27s |
| UDC-$x_{50}(\alpha=1)$ | 128.3373 | 0.02% | 14s | 258.1838 | 0.03% | 24s |
| UDC-$x_{250}(\alpha=1)$ | 128.3583 | 0.01% | 24s | 258.2236 | 0.01% | 56s |

| Method | KP2,000,$W$=200 | | | KP5,000,$W$=500 | | |
|---|---|---|---|---|---|---|
| | Obj.↓ | Gap | Time | Obj.↓ | Gap | Time |
| OR-Tools | 518.2880 | - | - | 1294.2263 | - | - |
| UDC-$x_{50}(\alpha=1)$ | 518.1653 | 0.02% | 8s | 1293.9041 | 0.02% | 16.2s |
| UDC-$x_{250}(\alpha=1)$ | 518.2337 | 0.01% | 24s | 1294.0966 | 0.01% | 40s |

**Experiment on min-max mTSP:** For min-max mTSP, we involve heuristic method hybrid genetic algorithm (HGA) [78] and LKH3, constructive method (parallel planning) DAN [46], Equity-Transformer [62] and DPN [42]. We use the dataset provided in DPN and the model of Equity-Transformer and DPN are finetuned on $N$=500, $M \in \{30, \ldots, 50\}$. The results in Table 12 demonstrate that the UDC can get outstanding results on CO problems with min-max objective functions as well.

Table 12: Objective function (Obj.), Gap to the best algorithm (Gap) on 500-node, 1,000-node min-max mTSP instances (100 instances). The best result is in bold.

| min-max mTSP | $N$=500,$M$=30 | | $N$=500,$M$=50 | | $N$=1,000,$M$=50 | |
|---|---|---|---|---|---|---|
| Methods | Obj.↓ | Gap | Obj.↓ | Gap | Obj.↓ | Gap |
| HGA | **2.0061** | - | 2.0061 | 0.00% | **2.0448** | - |
| LKH | 2.0061 | 0.00% | 2.0061 | 0.00% | 2.0448 | 0.00% |
| DAN | 2.2345 | 11.39% | 2.1465 | 7.00% | 2.3390 | 14.39% |
| Equity-Transformer-Finetune | 2.0165 | 0.52% | 2.0068 | 0.04% | 2.0634 | 0.91% |
| DPN-Finetune | 2.0065 | 0.02% | 2.0061 | 0.00% | 2.0452 | 0.02% |
| UDC-$\boldsymbol{x}_{50}(\alpha$=1) | 2.0840 | 3.89% | 2.1060 | 4.99% | 2.1762 | 6.43% |
| UDC-$\boldsymbol{x}_{50}(\alpha$=50) | 2.0087 | 0.13% | **2.0060** | - | 2.0495 | 0.23% |

## D.3 Experiments on TSP & CVRP Instances with Various Distributions

The cross-distribution generalization ability of NCO solvers is also necessary. So we evaluate the UDC on the Rotation distribution, and the Explosion distribution provided in Omni_VRP [79] on TSP and CVRP. The result is shown in Table 13, UDC demonstrates outstanding robustness on large-scale CO problems with different distributions.

Table 13: Experimental results on cross-distribution generalization. All four test sets are obtained from Omni_VRP [79] and contain 128 instances. The runtime marked with an asterisk (*) is proportionally adjusted (128/1,000) to match the size of our test datasets. The best learning-based method is marked by shade.

| | TSP1,000, Rotation | | | TSP1,000, Explosion | | |
|---|---|---|---|---|---|---|
| Method | Obj.↓ | Gap | Time | Obj.↓ | Gap | Time |
| Optimal | 17.2 | 0.00% | - | 15.63 | 0.00% | - |
| POMO | 24.58 | 42.84% | 8.5m | 22.7 | 45.24% | 8.5m |
| Omni_VRP+FS* | 19.53 | 14.30% | 49.9m | 17.75 | 13.38% | 49.9m |
| ELG | 19.09 | 10.97% | 15.6m | 17.37 | 11.16% | 13.7m |
| UDC-$\boldsymbol{x}_{250}(\alpha$=1) | 18.19 | 5.73% | 61s | 16.76 | 7.21% | 61s |

| | CVRP1,000, Rotation | | | CVRP1,000, Explosion | | |
|---|---|---|---|---|---|---|
| Method | Obj.↓ | Gap | Time | Obj.↓ | Gap | Time |
| Optimal | 32.49 | 0.00% | - | 32.31 | 0.00% | - |
| POMO | 64.22 | 97.64% | 10.2m | 59.52 | 84.24% | 11.0m |
| Omni_VRP+FS* | 35.6 | 10.26% | 56.8m | 35.25 | 10.45% | 56.8m |
| ELG | 37.04 | 14.00% | 16.3m | 36.48 | 12.92% | 16.6m |
| UDC-$\boldsymbol{x}_{250}(\alpha$=1) | 34.91 | 7.45% | 3.3m | 34.75 | 7.55% | 3.3m |

## D.4 Experiments on MIS Instances with Various Distributions

In addition to the TSP and CVRP, we conduct generalization experiments on the MIS problems. We conduct zero-shot generalization on the model trained by ER graphs with $N \sim U(700, 800)$ and $p$=0.15 to five other datasets, including:

- A random ER graph dataset with $N$=720 and $p$=0.05. (average 12,960 undirected edges)
- A random ER graph dataset with $N$=720 and $p$=0.25. (average 64,800 undirected edges)
- A random ER graph dataset with $N$=2,000 and $p$=0.05. (average 100,000 undirected edges)
- The ego-Facebook data in Stanford Large Network Dataset Collection (SNAP) (`https://snap.stanford.edu/data/`). (4,039 nodes, 88,234 undirected edges)
- The feather-lastfm-social data in SNAP. (7,624 nodes, 27,806 undirected edges)

All the first three ER random datasets contain 128 graph instances, and we consider the last two data as a single graph. The table below exhibits the comparison results between UDC and the SOTA heuristics KaMIS (implemented on `https://github.com/KarlsruheMIS/KaMIS`) on the original in-domain dataset (ER-[700-800], $p$=0.15) and the five out-of-domain datasets mentioned above. Obj. represents the objective function value, Gap represents the performance gap compared to the best algorithm and Time is the time consumption for solving the whole dataset.

Table 14: Objective function (Obj.), Gap to the best algorithm (Gap) on 500-node, 1,000-node min-max mTSP instances (100 instances). For MIS, the larger objective value (Obj.) is better.

| Dataset | ER, $N$=720, $p$=0.05 | | | ER-[700-800], $p$=0.15 (in-domain) | | |
|---|---|---|---|---|---|---|
| Method | Obj.↑ | Gap | Time | Obj.↑ | Gap | Time |
| KaMIS | 99.5 | - | 40.6m | 44.9 | - | 52.13m |
| UDC-$x_0(\alpha = 50)$ | 79.5 | 20.10% | 8s | 41.0 | 8.62% | 40s |
| UDC-$x_{50}(\alpha = 50)$ | 89.5 | 10.07% | 9.03m | 42.9 | 4.44% | 21.05m |
| UDC-$x_{250}(\alpha = 50)$ | 94.5 | 5.03% | 44.84m | 43.8 | 2.41% | 1.73h |

| Dataset | ER, $N$=720, $p$=0.25 | | | ER, $N$=2,000, $p$=0.05 | | |
|---|---|---|---|---|---|---|
| Method | Obj.↑ | Gap | Time | Obj.↑ | Gap | Time |
| KaMIS | 28.2 | - | 1.4h | 133.9 | - | 3h |
| UDC-$x_0(\alpha = 20)$ | 21.6 | 23.46% | 58s | 116.9 | 12.72% | 2m |
| UDC-$x_{50}(\alpha = 20)$ | 25.3 | 10.54% | 21m | 119.1 | 11.05% | 21m |
| UDC-$x_{250}(\alpha = 20)$ | 26.4 | 6.59% | 63m | 122.9 | 8.25% | 90m |

| Dataset | SNAP-ego-Facebook | | | SNAP-feather-lastfm-social | | |
|---|---|---|---|---|---|---|
| Method | Obj.↑ | Gap | Time | Obj.↑ | Gap | Time |
| KaMIS | 1052.0 | - | 155s | 4177.0 | - | 0.1s |
| UDC-$x_0(\alpha = 1)$ | 767.0 | 27.09% | 6s | 3622.0 | 13.29% | 2s |
| UDC-$x_{250}(\alpha = 1)$ | 901.0 | 14.35% | 11s | 3751.0 | 10.20% | 13s |
| UDC-$x_{2500}(\alpha = 1)$ | 1009.0 | 4.09% | 60s | 4067.0 | 2.63% | 80s |

Compared to KaMIS, UDC variants exhibit relatively stable performance gaps across datasets with various node sizes (n), sparsity (p), and generation methods, demonstrating good generalization ability. Moreover, except for the extremely sparse graph SNAP-feather-lastfm-social, UDC variants have significant advantages over KaMIS in terms of time.

Unfortunately, we are unable to re-implement any learning-based methods for MIS (e.g. Difusco [39] and T2T [66]), so we are unable to compare the generalization ability with other learning based algorithms. However, the stable performance gaps on different datasets partially demonstrate certain out-of-domain generalization performance of UDC on MIS.

## D.5 Experiments on Very Large-scale

This section evaluates the proposed UDC on very large-scale instances (i.e., 5,000-node, 7,000-node, and 10,000-node ones). On 5,000-node and 7,000-node CVRP, the proposed UDC-$x_{250}(\alpha=1)$ variant still demonstrate better performance.

However, on very large-scale TSP (results shown in Table 16), the UDC fails to converge to an outstanding performance in testing. We find that this is mainly due to a significant decrease in effectiveness when generalizing the dividing policy to 10,000-node TSP instances. So, we substitute the dividing policy to the random insertion used in [13], as shown in Table 17, The UDC with random-insertion-based initial solution achieved outstanding results. The experimental results preliminarily verify that in some problems such as TSP, UDC may generate unreliable dividing policies when processing large-scale generalization. In this case, using other initial solutions with a similar shape to the optimal solution can also obtain high-quality solutions.

Table 15: Objective function (Obj.), Gap to the best algorithm (Gap) on 5,000-node, 7,000-node CVRP (100 instances) [12, 13]. The best result is in bold and the best learning-based method is marked in shade.

| | CVRP5,000 | | | CVRP7,000 | | |
|---|---|---|---|---|---|---|
| Methods | Obj.↓ | Gap | Time | Obj.↓ | Gap | Time |
| LKH3 | 175.7 | 28.53% | 4.2h | 245.0 | 30.17% | 14h |
| BQ | 139.8 | 2.32% | 45m | - | - | - |
| LEHD | 138.2 | 1.09% | 3h | - | - | - |
| ICAM | 136.9 | 0.19% | 50m | - | - | - |
| TAM(LKH3)* | 144.6 | 5.80% | 35m | 196.9 | 4.61% | 1h |
| TAM(HGS)* | 142.8 | 4.48% | 1h | 193.6 | 2.86% | 1.5h |
| GLOP-G (LKH-3) | 140.4 | 2.69% | 8m | 191.2 | 1.59% | 10m |
| UDC-$x_2(\alpha=1)$ | 167.1 | 22.27% | 8m | 248.9 | 32.22% | 8m |
| UDC-$x_{50}(\alpha=1)$ | 140.8 | 2.99% | 9m | 195.8 | 4.03% | 13m |
| UDC-$x_{250}(\alpha=1)$ | **136.7** | - | 16m | **188.2** | - | 21.4m |

Table 16: Objective function (Obj.), Gap to the best algorithm (Gap) on 5,000-node (128 instances), 10,000-node TSP instances (16 instances). The best result is in bold and the best learning-based method is marked in shade.

| | TSP5,000 | | | TSP10,000 | | |
|---|---|---|---|---|---|---|
| Methods | Obj.↓ | Gap | Time | Obj.↓ | Gap | Time |
| LKH3 | **50.9** | - | 12m | **71.8** | - | 1h |
| BQ | 58.1 | 14.10% | 8m | - | - | - |
| LEHD | 59.2 | 16.23% | 20m | - | - | - |
| ICAM | 60.3 | 18.34% | 8m | - | - | - |
| GLOP-more revision | 53.3 | 4.54% | 21m | 75.3 | 4.90% | 1.8m |
| SO-mixed* | 52.6 | 3.25% | 55m | 74.3 | 3.52% | 2h |
| H-TSP | 55.0 | 7.99% | 26s | 77.8 | 8.33% | 50s |
| UDC-$x_2(\alpha=1)$ | 56.2 | 10.28% | 1.6m | 82.1 | 14.35% | 7m |
| UDC-$x_{50}(\alpha=1)$ | 54.5 | 7.06% | 2.6m | 79.2 | 10.41% | 7m |

Table 17: Objective function (Obj.), Gap to the best algorithm (Gap) on 10,000-node, 100,000-node TSP instances (16 instances). The best result is in bold and the best learning-based method is marked in shade. We use the TSP100,000 result reported in GLOP [13] on probably a different test set.

| | TSP10,000 | | | TSP100,000 | | |
|---|---|---|---|---|---|---|
| Methods | Obj.↓ | Gap | Time | Obj.↓ | Gap | Time |
| LKH3 | **71.8** | - | 1h | **226.4*** | - | 8.1h |
| DIMES+S | 86.3 | 20.18% | 3.1m | 286.1* | 26.37%* | 2.0m |
| GLOP-more revision | 75.3 | 4.90% | 1.8m | 238.0* | 5.12%* | 2.8m |
| UDC-$x_{50}(\alpha=1)$ | 74.7 | 4.11% | 18s | 235.8 | 4.15% | 3.5m |
| UDC-$x_{250}(\alpha=1)$ | 74.7 | 4.03% | 1m | 235.6 | 4.05% | 11.5m |

# E Ablation Study

## E.1 Ablation Study on $r$ (Convergence Speed In Testing)

The proposed UDC adopts the multi-conquering stage ($r$ stages) in common variants. Therefore, as an iterative method, the effectiveness of our testing phase is closely related to the testing time. We record the iterative convergence curves of the objective function value (Obj.) during testing TSP1,000, CVRP1,000, and PCTSP1,000 with UDC($\alpha$=1), UDC($\alpha$=50), GLOP [13], LEHD (RRC) [8], and DIFUSCO [39]. The results are shown in Figure 5, and curves (except for DIFUSCO, which uses reported results) are generated under the same test set and computation resources. We can obtain the following conclusions:

1. For the two variants of UDC, UDC($\alpha$=1) can generate better solutions compared to UDC($\alpha$=50) in a limited period but will trap into local optima quickly. For PCTSP, there is no significant difference in the final convergence result between the two versions.

2. Compared to the advanced neural divide-and-conquer method GLOP, the objective functions of both the two UDC variants always significantly lead when going through the same testing time.

3. Compared with the search-enabled heatmap-based method DIFUSCO, although the heatmap-based method can ultimately obtain a superior value of the objective function through a sufficient search on TSP, the two variants of UDC lead in the value of the objective function in a finite period.

4. Compared to the SL-based sub-path solver LEHD with RRC [8], the RL-based UDC($\alpha$=50) variant demonstrates competitive objective function values on CVRP, while both the two variants of UDC show higher efficiency on TSP and CVRP.

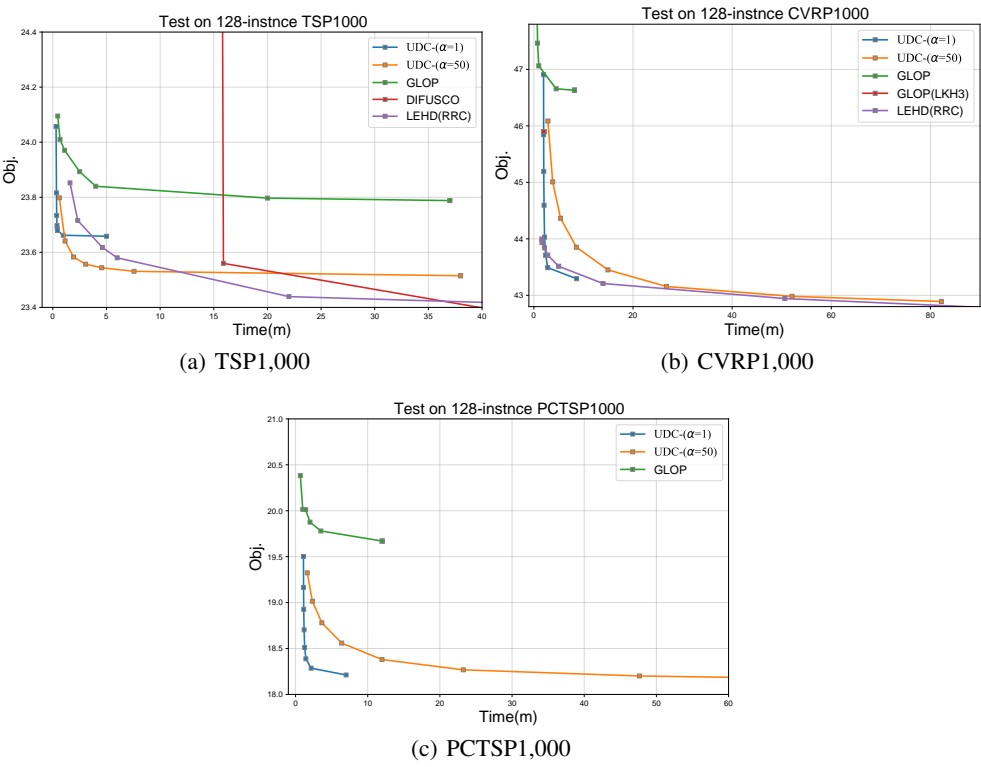

Figure 5: Testing curves for ablation study.

## E.2 Ablation Study on $\alpha$ (Sampled Initial Solutions)

In this subsection, we demonstrate the change in effectiveness of UDC relating to the numbers of initial solutions (i.e., $\alpha$) for each CO problem. We use different $\alpha$ values to test the TSP, CVRP,

OP, PCTSP, and SPCTSP (CO problems evaluated in the main paper) on their original test set. As exhibited in Table 18, results with $\alpha = 50$ always demonstrate significantly better performance but much more time consumption.

However, the sensitivity of $\alpha$ on performance varies among different CO problems. In TSP, CVRP, and OP, the differences between results with $\alpha = 50$ and results with $\alpha = 1$ appear to be quite significant, while in PCTSP and SPCTSP, employing more initial solutions demonstrates no significant effect. Moreover, On 2,000-node problems TSP, CVRP, and OP, the effect of $\alpha = 50$ is particularly pronounced. Since the difference in initial solutions is entirely from sampling the dividing policy, a more significant distinction between $\alpha = 50$ and $\alpha = 1$ variants might mean that the initial solution prediction is more difficult in these instances. Meanwhile, the more significant difference in 2,000-node instances indicates that compared to (S)PCTSP, the dividing policy has worse scaling ability on TSP, CVRP, and OP.

Table 18: Ablation on the number of initial solution $r$. Objective function values (Obj.), Gap to the best algorithm (Gap) on 500-node, 1,000-node, and 2,000-node instances are listed.

| | TSP500 | | | TSP1,000 | | | TSP2,000 | | |
|---|---|---|---|---|---|---|---|---|---|
| Methods | Obj.↓ | Gap | Time | Obj.↓ | Gap | Time | Obj.↓ | Gap | Time |
| LKH3 | 16.52 | - | 5.5m | 23.12 | - | 24m | 32.45 | - | 1h |
| UDC-$x_2(\alpha{=}1)$ | 17.12 | 3.62% | 10s | 24.06 | 4.05% | 18s | 34.44 | 6.13% | 48s |
| UDC-$x_{50}(\alpha{=}1)$ | 16.92 | 2.41% | 14s | 23.68 | 2.42% | 26s | 33.60 | 3.54% | 1.1m |
| UDC-$x_2(\alpha{=}50)$ | 16.94 | 2.54% | 20s | 23.79 | 2.92% | 32s | 34.14 | 5.19% | 23s |
| UDC-$x_{50}(\alpha{=}50)$ | 16.78 | 1.58% | 4m | 23.53 | 1.78% | 8m | 33.26 | 2.49% | 15m |

| | CVRP500 | | | CVRP1,000 | | | CVRP2,000 | | |
|---|---|---|---|---|---|---|---|---|---|
| Methods | Obj.↓ | Gap | Time | Obj.↓ | Gap | Time | Obj.↓ | Gap | Time |
| LKH3 | 37.23 | - | 7.1h | 46.40 | 8.18% | 14m | 64.90 | 8.14% | 20min |
| UDC-$x_2(\alpha{=}1)$ | 40.64 | 9.15% | 1m | 46.91 | 9.09% | 2m | 68.86 | 14.74% | 3.1m |
| UDC-$x_{50}(\alpha{=}1)$ | 38.68 | 3.89% | 1.2m | 43.98 | 2.28% | 2.2m | 62.10 | 3.48% | 3.5m |
| UDC-$x_{250}(\alpha{=}1)$ | 38.32 | 2.93% | 2.1m | 43.46 | 1.06% | 3.4m | 61.03 | 1.70% | 5.5m |
| UDC-$x_2(\alpha{=}50)$ | 40.05 | 7.57% | 1m | 46.09 | 7.18% | 2.1m | 65.26 | 8.75% | 5m |
| UDC-$x_{50}(\alpha{=}50)$ | 38.35 | 3.01% | 9.6m | 43.48 | 1.11% | 14m | 60.94 | 1.55% | 23m |
| UDC-$x_{250}(\alpha{=}50)$ | 38.00 | 2.06% | 1.1h | 43.00 | - | 1.1h | 60.01 | - | 2.15h |

| | OP500 | | | OP1,000 | | | OP2,000 | | |
|---|---|---|---|---|---|---|---|---|---|
| Methods | Obj.↑ | Gap | Time | Obj.↑ | Gap | Time | Obj.↑ | Gap | Time |
| EA4OP | 81.70 | - | 7.4m | 117.70 | - | 63.4m | 167.74 | - | 38.4m |
| UDC-$x_2(\alpha{=}1)$ | 75.28 | 7.86% | 17s | 109.42 | 7.03% | 25s | 135.34 | 19.32% | 5s |
| UDC-$x_{50}(\alpha{=}1)$ | 78.02 | 4.51% | 23s | 111.72 | 5.08% | 23s | 139.25 | 16.99% | 8s |
| UDC-$x_{250}(\alpha{=}1)$ | 79.27 | 2.98% | 48s | 113.27 | 3.76% | 1.1m | 144.10 | 14.09% | 21s |
| UDC-$x_2(\alpha{=}50)$ | 75.61 | 7.46% | 25s | 110.25 | 6.33% | 40s | 147.48 | 12.08% | 10s |
| UDC-$x_{50}(\alpha{=}50)$ | 78.55 | 3.86% | 5m | 112.59 | 4.34% | 9.5m | 151.89 | 9.45% | 2m |
| UDC-$x_{250}(\alpha{=}50)$ | 79.81 | 2.32% | 24m | 114.16 | 3.01% | 47m | 159.10 | 5.15% | 11m |

| | PCTSP500 | | | PCTSP1,000 | | | PCTSP2,000 | | |
|---|---|---|---|---|---|---|---|---|---|
| Methods | Obj.↓ | Gap | Time | Obj.↓ | Gap | Time | Obj.↓ | Gap | Time |
| OR-Tools* | 14.40 | 10.19% | 16h | 20.60 | 13.34% | 16h | - | - | - |
| UDC-$x_2(\alpha{=}1)$ | 13.95 | 6.78% | 34s | 19.50 | 7.30% | 1m | 27.22 | 8.87% | 18s |
| UDC-$x_{50}(\alpha{=}1)$ | 13.25 | 1.39% | 42s | 18.46 | 1.59% | 1.3m | 25.46 | 1.81% | 23s |
| UDC-$x_{250}(\alpha{=}1)$ | 13.17 | 0.74% | 80s | 18.29 | 0.61% | | 25.04 | 0.12% | 53s |
| UDC-$x_2(\alpha{=}50)$ | 13.80 | 5.61% | 40s | 19.32 | 6.32% | 1.6m | 26.76 | 7.02% | 24s |
| UDC-$x_{50}(\alpha{=}50)$ | 13.15 | 0.63% | 7m | 18.33 | 0.86% | 15m | 25.22 | 0.85% | 3.4m |
| UDC-$x_{250}(\alpha{=}50)$ | 13.07 | - | 40m | 18.18 | - | 70m | 25.01 | - | 17.6m |

| | SPCTSP500 | | | SPCTSP1,000 | | | SPCTSP2,000 | | |
|---|---|---|---|---|---|---|---|---|---|
| Methods | Obj.↓ | Gap | Time | Obj.↓ | Gap | Time | Obj.↓ | Gap | Time |
| UDC-$x_2(\alpha{=}1)$ | 14.21 | 8.20% | 42s | 20.01 | 9.04% | 1.1m | 28.15 | 11.15% | 20s |
| UDC-$x_{50}(\alpha{=}1)$ | 13.40 | 2.04% | 1m | 18.93 | 3.11% | 1.4m | 26.24 | 3.60% | 29s |
| UDC-$x_{250}(\alpha{=}1)$ | 13.24 | 0.82% | 2.3m | 18.54 | 1.03% | 2.6m | 25.38 | 0.22% | 1.1m |
| UDC-$x_2(\alpha{=}50)$ | 14.05 | 6.98% | 40s | 19.74 | 7.52% | 1.6m | 27.98 | 10.48% | 24s |
| UDC-$x_{50}(\alpha{=}50)$ | 13.27 | 1.01% | 7m | 18.71 | 1.91% | 15m | 26.17 | 3.34% | 3.4m |
| UDC-$x_{250}(\alpha{=}50)$ | 13.13 | - | 40m | 18.35 | - | 70m | 25.33 | - | 17.6m |

### E.3 Ablation Study on the Generation Method of Initial Solutions

To check the ability of the dividing policy, we further conduct experiments on TSP with different generation methods of initial solutions, and the results shown in Table 19 demonstrate the significance of a good initial solution to the final performance. The UDC variants with a random initial solution or a nearest greedy initial solution cannot converge to good objective values even with 50 conquering stages. Moreover, from TSP500 to TSP2,000, the dividing policy of UDC produces similar results to the heuristic algorithm random insertion (used in GLOP [13]), which verifies the quality of the dividing policy.

Table 19: Ablation on different algorithms as initial solutions $x_0$. The conquering stages of all the variants are the constructive of UDC. Random-UDC directly represents a random solution as the initial solution. Nearest Greedy-UDC uses the nearest greedy algorithm (starting from the first node) and Random Insertion-UDC employs the random insertion heuristic as the initial solution. $\alpha = 1$ in all UDC variants. UDC-$x_2$ and UDC-$x_{50}$ is the original version. Obj. is the objective value and Gap represents the gap to the best method.

| Method | TSP500 | | TSP1,000 | | TSP2,000 | |
| --- | --- | --- | --- | --- | --- | --- |
| Method | Obj. | Gap | Obj. | Gap | Obj. | Gap |
| Concorde | 16.52 | - | 23.12 | - | 32.45 | - |
| Random-UDC-$x_2$ | 33.98 | 105.68% | 67.82 | 193.34% | 134.98 | 315.98% |
| Random-UDC-$x_{50}$ | 26.11 | 58.01% | 52.48 | 126.98% | 104.48 | 221.96% |
| Nearest Greedy-UDC-$x_2$ | 18.48 | 11.87% | 26.42 | 14.27% | 37.48 | 15.50% |
| Nearest Greedy-UDC-$x_{50}$ | 17.88 | 8.23% | 25.75 | 11.38% | 36.80 | 13.40% |
| Random Insertion-UDC-$x_2$ | 17.09 | 3.45% | 24.08 | 4.17% | 34.00 | 4.78% |
| Random Insertion-UDC-$x_{50}$ | 16.84 | 1.93% | 23.75 | 2.73% | 33.55 | 3.38% |
| UDC-$x_2(\alpha = 1)$ | 17.12 | 3.62% | 24.06 | 4.05% | 34.44 | 6.13% |
| UDC-$x_{50}(\alpha = 1)$ | 16.92 | 2.41% | 23.68 | 2.42% | 33.60 | 3.54% |

### E.4 Ablation Study on Components

In the main part, we have shown the result of the ablation study on the unified training scheme and the conquering policy, demonstrating the superiority of the unified training scheme and the flexibility in selecting a model for the conquering policy.

In this sub-section, we conduct experiments to demonstrate the necessity of three other components of UDC, including the Reunion step in the DCR training method, and the coordinate transformation& constraint normalization in the conquering stage.

Figure 6(a)(b) compares the training efficiency of the Original UDC to the UDC variant without the Reunion step in DCR (i.e., the w/o Reunion in the DCR variant) and the UDC variant without the coordinate transformation (i.e., normalization) in the conquering stage (i.e., the w/o Norm in the Conquering Stage). The training curve on both TSP500 and TSP 1,000 demonstrates the significance of both two components and the Reunion step seems more significant for a better training convergency.

Moreover, we also evaluate the significance of constraint normalization on PCTSP. As shown in Figure 6(c), The final performance after 250 epochs and training efficiency of UDC is strongly nerfed without the normalization on the total prize lower-bound in sub-(S)PCTSPs.

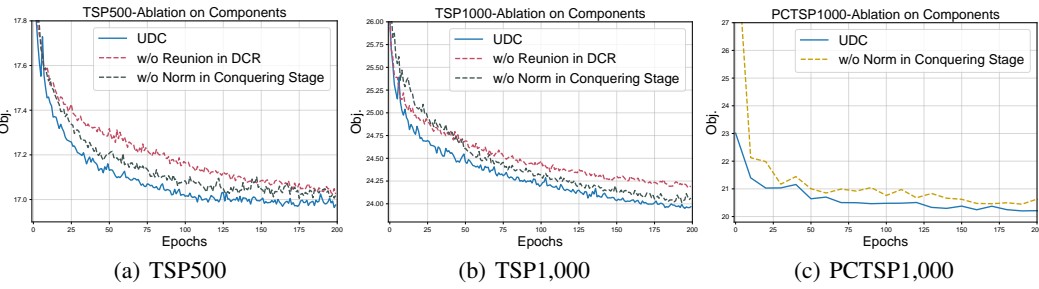

Figure 6: Training curves for ablation study.

### E.5 Ablation Study on Disable DCR in KP

For KP, there will be no sub-optimal connection as shown in Figure 2, so there is no logical difference in whether DCR is enabled or not. We provide the KP results of Enable DCR in Table 20 and the results support the above conclusion.

Table 20: Ablation on whether to enable the DCR in training UDC for KP. The version with disabling DCR (i.e., Disable DCR) is the same as Table 11 in the original paper and the datasets are also the same. The value shown in the table is the gap to optimal (OR-Tools).

| Method | KP500 | KP1,000 | KP2,000 | KP5,000 |
|---|---|---|---|---|
| UDC-$x_{50}$(Disable DCR) | 0.0247% | 0.0260% | 0.0237% | 0.0249% |
| UDC-$x_{250}$(Disable DCR) | 0.0083% | 0.0106% | 0.0105% | 0.0100% |
| UDC-$x_{50}$(Enable DCR) | 0.0105% | 0.0258% | 0.0254% | 0.0273% |
| UDC-$x_{250}$(Enable DCR) | 0.0080% | 0.0117% | 0.0128% | 0.0095% |

### E.6 Supplementary Ablation Study on Conquering Policies of KP

UDC exhibits flexibility in choosing a constructive solver for conquering policy. For well-developed CO problems like TSP and KP, there are multiple constructive solvers available. Section 5.2 presents an ablation study on the selection of these constructive solvers for TSP. In this subsection, we conducted a similar ablation study on KP, and the results are displayed in Table 21. Both sets of results indicate that the final performance of UDC is not sensitive to the model selection for the conquering policy.

Therefore, when generalizing UDC to new CO problems, it may not be necessary to specifically choose which available constructive solver to model the conquering policy.

Table 21: Ablation on the choice of constructive model as the conquering policy in KP. ICAM and POMO are two available constructive solvers for KP. The datasets are the same as Table 11 in the original paper. The value shown in the table is the gap to optimal (OR-Tools).

| Method | KP500 | KP1,000 | KP2,000 | KP5,000 |
|---|---|---|---|---|
| UDC-$x_{50}$(ICAM) | 0.0247% | 0.0260% | 0.0237% | 0.0249% |
| UDC-$x_{250}$(ICAM) | 0.0083% | 0.0106% | 0.0105% | 0.0100% |
| UDC-$x_{50}$(POMO) | 0.0255% | 0.0252% | 0.0253% | 0.0242% |
| UDC-$x_{250}$(POMO) | 0.0098% | 0.0107% | 0.0119% | 0.0113% |

# F Visualization

In this section, we provide visualization to explain our procedure and show our effectiveness directly. As shown in 7, 8, and 9, respectively. Figures exhibit the initial solution, optimal solution, and reported solutions (i.e., UDC-$x_2(\alpha=50)$, UDC-$x_{50}(\alpha=50)$, and UDC-$x_{250}(\alpha=50)$). The singular shape of initial solutions verifies that the dividing policy is mainly not to obtain a shorter predicted path, but to predict an initial solution that is easier to improve to the optimal solution by the following two conquering stages. In addition, these examples also preliminarily verify that our trained UDC can handle large-scale TSP, large-scale CVRP, and large-scale OP well after several rounds of conquering.

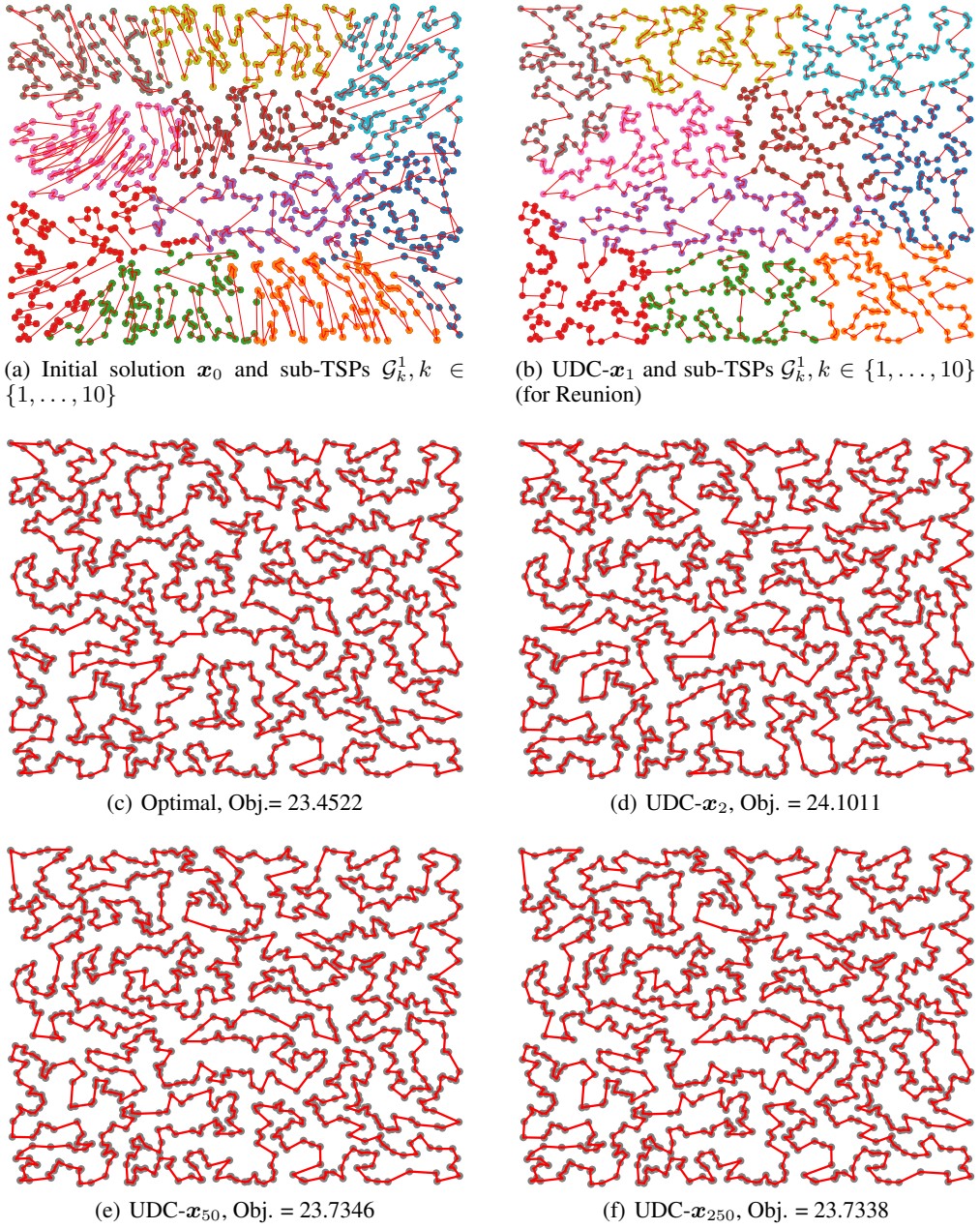

(a) Initial solution $x_0$ and sub-TSPs $\mathcal{G}_k^1, k \in \{1, \ldots, 10\}$

(b) UDC-$x_1$ and sub-TSPs $\mathcal{G}_k^1, k \in \{1, \ldots, 10\}$ (for Reunion)

(c) Optimal, Obj.= 23.4522

(d) UDC-$x_2$, Obj. = 24.1011

(e) UDC-$x_{50}$, Obj. = 23.7346

(f) UDC-$x_{250}$, Obj. = 23.7338

Figure 7: Visualization of TSP solutions on a random TSP instance, Obj. represents the objective function. The colors in subfigure (a)(b) represent sub-problems.

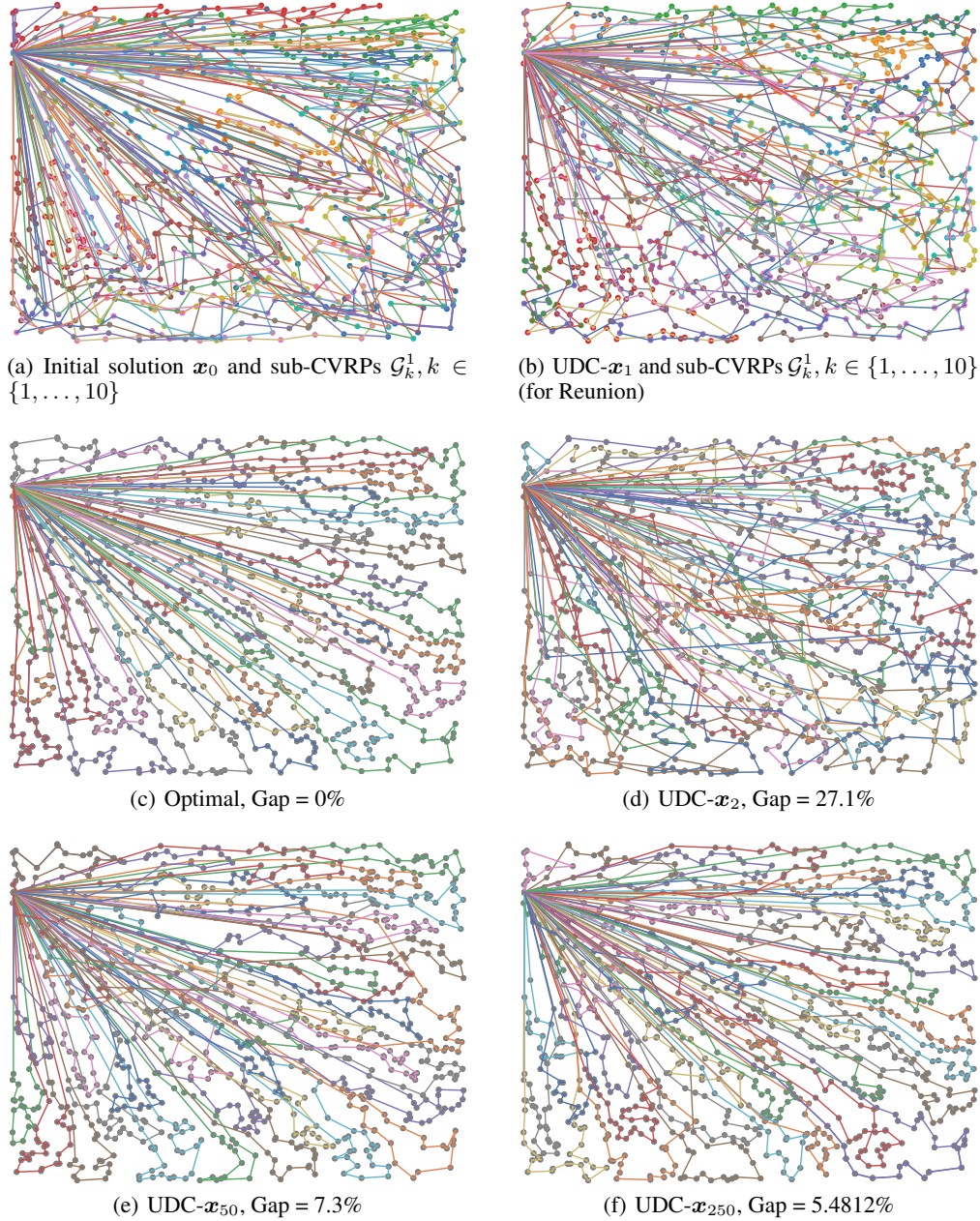

(a) Initial solution $\boldsymbol{x}_0$ and sub-CVRPs $\mathcal{G}_k^1, k \in \{1, \dots, 10\}$

(b) UDC-$\boldsymbol{x}_1$ and sub-CVRPs $\mathcal{G}_k^1, k \in \{1, \dots, 10\}$ (for Reunion)

(c) Optimal, Gap = 0%

(d) UDC-$\boldsymbol{x}_2$, Gap = 27.1%

(e) UDC-$\boldsymbol{x}_{50}$, Gap = 7.3%

(f) UDC-$\boldsymbol{x}_{250}$, Gap = 5.4812%

Figure 8: Visualization of CVRP solutions on a SetX CVRPLib instance X-n1001-k43, Gap represents the gap of objective function compared to optimal. The colors in subfigure (a)(b) represent sub-problems.

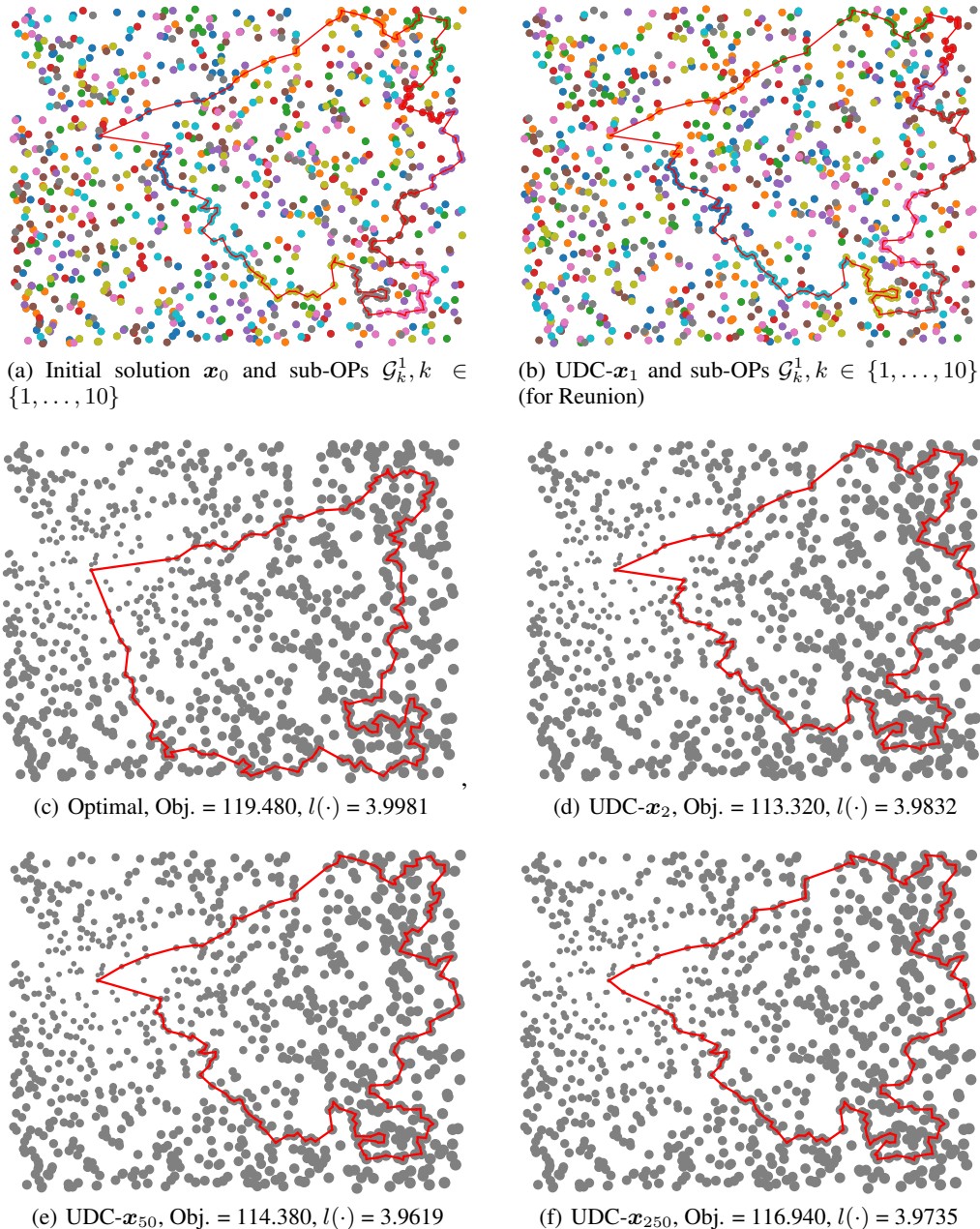

(a) Initial solution $\boldsymbol{x}_0$ and sub-OPs $\mathcal{G}_k^1, k \in \{1, \ldots, 10\}$

(b) UDC-$\boldsymbol{x}_1$ and sub-OPs $\mathcal{G}_k^1, k \in \{1, \ldots, 10\}$ (for Reunion)

(c) Optimal, Obj. = 119.480, $l(\cdot)$ = 3.9981

(d) UDC-$\boldsymbol{x}_2$, Obj. = 113.320, $l(\cdot)$ = 3.9832

(e) UDC-$\boldsymbol{x}_{50}$, Obj. = 114.380, $l(\cdot)$ = 3.9619

(f) UDC-$\boldsymbol{x}_{250}$, Obj. = 116.940, $l(\cdot)$ = 3.9735

Figure 9: Visualization of OP solutions on a random OP instance, the size of nodes from sub-figure (c) to (f) represents their values $\rho_i$. Obj. represents the objective function and $l(\cdot)$ is the length of the solution defined in Eq. (10). The colors in subfigure (a)(b) represent sub-problems.

# G Baselines & License

**Classical solvers.** For LKH3, We adopt the commonly used setting in NCO methods [51] and implement the LKH3 in TSP, CVRP, OVRP, and min-max mTSP based on the code from `https://github.com/wouterkool/attention-learn-to-route`. For min-max mTSP, the parameter MAX_CANDIDATES is set to 6, and for OVRP VEHICLE is set to 100. In mim-max mTSP, the implemented HGA algorithm is given in `https://github.com/Sasanm88/m-TSP`. For OR-Tools, we implement the code in `https://developers.google.com/optimization/pack/knapsack?hl=zh-cn` for KP. EA4OP `https://github.com/gkobeaga/op-solver` is also implemented as an OP baseline and KaMIS `https://github.com/KarlsruheMIS/KaMIS` is implemented for MIS.

**SL-based sub-path solvers** SL-based sub-path solvers contain the LEHD [8] and BQ [14]. In addressing OP by BQ, we fail to get legal solutions so we use the reported gap to EA4OP.

**RL-based constructive solvers** This paper involves AM [51], POMO [5], ELG [12], ICAM [16], MDAM [52], MatNet [41], Omni_VRP [79], and Sym-NCO [63] as baselines. Constructive solver DAN [46], DPN [42] and Equity-Transformer [62] are employed for min-max mTSP, and MatNet is specially employed for ATSP. For POMO, ELG, Omni_VRP, and ICAM, We use the **multi-start with x8 augmentation setting** for CO instances with size $N \leq 1,000$ and multi-start with no augmentation for instances with $N > 1,000$ [16]. Specifically, the ELG aug×8 in Table 4 adopts a special setting given in its paper [80]. We adopt 200-augmentation for Sym-NCO on CO problems with size $N \leq 1,000$ and 100-augmentation on 2,000-node CO problems.

**Heatmap-based** We fail to implement all involved heatmap-based solvers (i.e., Attn-MCTS [37], DIMES [35], DIFUSCO [39], and T2T [66]), so we adopt the report results for comparison.

**Neural divide-and-conquer methods** We implement H-TSP [20] and GLOP [13] as baselines. For L2D, TAM, and SO, we list their reported results in comparisons. The random insertion in 17 is generated based on code in GLOP `https://github.com/henry-yeh/GLOP`

Considering **datasets**, we use the benchmark TSPLib, CVRPLib, two SNAP graphs, and two provided test sets in DIMES `https://github.com/DIMESTeam/DIMES?utm_source=catalyzex.com` (MIS) and Omni_VRP `https://github.com/RoyalSkye/Omni-VRP` (TSP & CVRP).

Table 22: A summary of licenses.

| Resources | Type | License |
|---|---|---|
| LKH3 | Code | Available for academic research use |
| HGA | Code | Available online |
| KaMIS | Code | MIT License |
| OR-Tools | Code | Apache License, Version 2.0 |
| LEHD | Code | Available for any non-commercial use |
| BQ | Code | CC BY-NC-SA 4.0 license |
| AM | Code | MIT License |
| POMO | Code | MIT License |
| ELG | Code | MIT License |
| MDAM | Code | MIT License |
| Sym-NCO | Code | Available online |
| MatNet | Code | MIT License |
| Omni_VRP | Code | MIT License |
| DAN | Code | MIT License |
| Equity-Transformer | Code | Available online |
| H-TSP | Code | Available for academic research use |
| GLOP | Code | MIT License |
| TSPLib | Dataset | Available for any non-commercial use |
| CVRPLib [54, 55] | Dataset | Available for any non-commercial use |
| Omni_VRP | Dataset | MIT License |
| DIMES | Dataset | MIT License |
| SNAP (ego-Facebook & feather-lastfm-social) | Dataset | Available online |

The licenses for codes and datasets used in this work are listed in Table 22.

