# OpenReview forum: "UDC: A Unified Neural Divide-and-Conquer Framework for Large-Scale Combinatorial Optimization Problems"
_NeurIPS.cc/2024/Conference — NeurIPS 2024 poster_

### Official Review · Reviewer_Uqww · 2024-07-06

**Soundness:** 3
**Presentation:** 3
**Contribution:** 4
**Rating:** 7
**Confidence:** 4

**Summary:**

UDC focuses on the divide-and-conquer-based NCO methods, proposing a novel framework that does not require heuristics and adopting an efficient training approach DCR. Compared to existing methods, UDC has significant improvements in effectiveness and applicability.

**Strengths:**

1. This paper conducts extensive experiments. UDC exhibits effectiveness in 10 different CO problems, improving the existing divide-and-conquer methods in both effectiveness and applicability.
2. This paper validates the importance of unified training. The idea of DCR is quite novel.
3. The article is easy to follow and experiments give detailed explanations on the setting reason for hyperparameters and components.

**Weaknesses:**

There are no major weaknesses in the article, my questions are presented in the Question section.

**Questions:**

1. UDC shows wide applicability compared to existing NCO methods. What is the reason for its significant advantage in applicability compared to existing divide-and-conquer methods?
2. The solution representation in the upper part of Figure 2 is not clear enough, please add illustrations in the Figure caption.
3. On TSPLib and CVRPLib, UDC seems to perform less prominently. Please provide the result of UDC on benchmark instances of other CO problems.
4. Please add more clear indexes in the main part for contents in the Appendix.
5. Should GLOP for TSPLib in Table 4 be GLOP more revision?
6. Do BS and bs in Table 3 represent beam search? Please provide a detailed introduction.

**Limitations:**

Chapter 3.3 discusses the unavailability of UDC on certain CO problems, such as TSPTW. This should also be included in discussing the limitation.

---

> ### Author Rebuttal · Authors · 2024-08-05
>
> Thank you very much for your time and effort in reviewing our work. We are glad to know that you find the idea of DCR quite novel, UDC exhibits effectiveness in 10 different CO problems, the article is easy to follow, and experiments give detailed explanations on the setting reason for hyperparameters and components.
>
> We address your concerns point-by-point as follows:
>
>
> >**Question 1. Reason for applicability**: UDC shows wide applicability compared to existing NCO methods. What is the reason for its significant advantage in applicability compared to existing divide-and-conquer methods?
>
>
> The wide application of UDC on large-scale CO problems mainly depends on two reasons as follows: **1)** UDC gets rid of the heuristic components in both the dividing policy and the conquering policy (as shown in Table 1 in the original paper). Heuristic components often require expert knowledge to redesign when handling unseen instances and new CO problems, thus undermining the applicability of methods. **2)** The efficient training method DCR enables a unified training framework for all CO problems. The unified training framework does not rely on the design of separate pre-training procedures or tunings for specific CO problems, making UDC easy to apply to new CO problems.
>
>
> >**Question 3. Benchmark experiments**: Please provide the result of UDC on benchmark instances of other CO problems.
>
> Thank you for your suggestion. We further evaluate the proposed UDC on the benchmark dataset of min-max mTSP. The benchmark dataset of min-max mTSP contains instances from 500 to 1,000 nodes. It includes the u574 instance, the p654 instance, and the rat783 instance (selected by the paper of Equity-Transformer [1] and we use the reported results from [1]). Objective values of methods are shown in the table below. Compared to Equity-Transformer and the heuristic method LKH-3 and OR-Tools, UDC-$\boldsymbol{x}_{50}$ ($\alpha=50$) can significantly lead.
>
> | Instance  | $M$= | LKH-3 | OR-Tools | Equity-Transformer | UDC-$\boldsymbol{x}_{50}$ ($\alpha=50$) |
> |-----------|------|-------|----------|---------------------|-----------------------------------------|
> |    | 30   | 8800  | 19391    | 6642                | 6641                                    |
> | u574      | 40   | 8051  | 15924    | 6642                | 6641                                    |
> |    | 50   | 7733  | 14192    | 6642                | 6641                                    |
> |    | 30   | 13317 | 25552    | 12795               | 12380                                   |
> |   p654      | 40   | 13668 | 25547    | 12795               | 12303                                   |
> |     | 50   | 13188 | 25547    | 12795               | 12270                                   |
> |   | 30   | 2217  | 5105     | 1272                | 1243                                    |
> | rat783    | 40   | 1872  | 5105     | 1272                | 1232                                    |
> |    | 50   | 1640  | 4005     | 1272                | 1232                                    |
>
>
> >**Question 2,4. Writing clarity.**
>
> Thanks for your valuable suggestion. The upper part of Figure 2 shows the pipeline of the training method DCR. In the DCR-enabled training process, there are two conquering stages, the Conquer step and the Reunion step, where the Reunion step is introduced to eliminate the negative impact of the sub-optimal dividing policy. The lower part provides a local view of a solution fragment to demonstrate our motivation. DCR has the potential to correct wrong dividing results by generating the solution of the connection part in the Reunion step.
>
> We will include this illustration in the caption of Figure 2, use more appropriate colors for better expression, and add more indexes in the main part for contents in the Appendix of the original manuscript.
>
> >**Question 5,6. Abbreviation in Tables.**
>
> Thank you for your comments. Yes, in Table 4 in the original paper, the GLOP for TSP should be GLOP more revision instead of GLOP-LKH3. We will change the index in Table 4 to GLOP and provide an additional description of the settings.
>
> Yes, both BS and bs in Table 3 of the original manuscript represent the beam search. Beam search is a commonly used decoding strategy in autoregressive sequence model decoding, which is a form of breadth-first search (BFS) [2]. Attention Model [3] first introduces the technique into the field of NCO. For a beam limited to k (noted as bs$k$ in Table 3 of the original manuscript, e.g., bs1024 or bs16), the beam search will construct k solutions with top probabilities sampled from the model policy. Models with beam search in decoding can get better results with more solving time.
>
>
> >**Limitation**: Chapter 3.3 discusses the unavailability of UDC on certain CO problems, such as TSPTW. This should also be included in discussing the limitation.
>
> Thank you for pointing this out, we will include the content discussed in Chapter 3.3 in the illustration on limitations.
>
> ***
>
> >**References**
>
> [1] Son, Jiwoo, et al. "Equity-Transformer: Solving NP-Hard Min-Max Routing Problems as Sequential Generation with Equity Context." AAAI 2024, 2024.
>
> [2] Meister, Clara, Tim Vieira, and Ryan Cotterell. "If beam search is the answer, what was the question?." arXiv preprint, 2020.
>
> [2] Kool, Wouter, Herke Van Hoof, and Max Welling. "Attention, learn to solve routing problems!." arXiv preprint, 2018.

---

> > ### Comment · Reviewer_Uqww · 2024-08-08
> >
> > Thank you for your response, which has addressed my concerns. I raised my score to 7. I appreciate the novelty and effectiveness of UDC, especially its wide applicability. This work is inspiring to the NCO research so I think it should be accepted.

---

> > > ### Author Response · Authors · 2024-08-08
> > >
> > > Thanks for your support and raising the score to 7. We are delighted to hear your appreciation for the novelty, effectiveness, and wide applicability of UDC.

---

### Official Review · Reviewer_kAJo · 2024-07-08

**Soundness:** 3
**Presentation:** 2
**Contribution:** 2
**Rating:** 5
**Confidence:** 4

**Summary:**

The paper introduces a novel Unified Neural Divide-and-Conquer (UDC) framework designed to address large-scale CO problems. The UDC framework leverages a Divide-Conquer-Reunion (DCR) training method that uses GNN for global instance division and a constructive neural solver for sub-problem solutions. This unified approach aims to improve both efficiency and applicability across various CO problems by combining the training process of the dividing stage and the conquering stage.

**Strengths:**

1. The DCR method proposes a view of unifying the training of dividing and conquering stages for solving large-scale CO problems.
2. The UDC framework is designed with applicability to a wider range of CO problems, without heavy reliance on problem-specific heuristics like in the previous works.
3. The idea of considering interdependencies between dividing and conquering stages is interesting as a neural CO training scheme.
4. The authors conduct extensive experiments where the proposed framework's performance is shown to be generally good on certain problems.

**Weaknesses:**

1. The novelty of the proposed framework requires further explanation. The current UDC/DCR seems to be a combination of existing techniques in either dividing or conquering stages. From a local perspective, 1) the sub-problem dividing policy is simply a conventional continuous node split based on the initial solution, 2) the conquering phase is a direct exploitation of existing problem-specific models like POMO, ICAM, MatNet, etc., to tackle the small scaled sub-problem, and 3) the technique in the reunion part resembles prevalent tricks in the neural CO literature, such as POMO/random-start/multi-sampling/random augmentation, etc. Therefore, the proposed unified framework is more of a mathematical or conceptual summarization of the current RL-based divide-and-conquer pipeline rather than a novel methodology for tackling various CO problems using a unified architecture.
2. The training procedure is not stated clearly enough. As two different networks are adopted in either phase, how are the gradients in equation (6) updated simultaneously during an instantiated epoch? If the initial solution is generated using a heatmap and some decoding scheme while the sub-solutions are computed auto-regressively with another model, how the REINFORCE algorithms are implemented necessitates further presentation in order to address obscurity regarding implementation details.
3. The solving time of UDC is relatively longer than several compared neural methods which is questionable of its efficiency and practiability. Further notably is the rapidly increased time consumed with the raising number of stages it takes in conquering, to which, on the other hand, the solving quality actually highly relates. Thus, further experiments shall be necessary to explicitly illustrate the relationship among solving quality, the number of conquering stages and solving time.
4. Will it benefit from using supervised learning over current RL method for sub-problem conquering? Note that SL-based approaches achieve better performance in problems like TSP in previous problem-specific research.
5. Can the framework generalize to a wider range of CO problems? Ten problems as the authors evaluated in this paper, most of them have a similar setting or formulation to routing tasks like TSP, VRP or their variants.
6. Experiments on different initial solutions are missing. To what extent are the effectiveness of the UDC framework rely on the quality of the initial solution generated during the dividing stage? Note the fact that a poor initial solution may lead to sub-optimal conquering solutions (as mentioned in Appendix D.4), more empirical studies or ablation should be conducted regarding the ways and quality of initial solution generation.
7. Unsatisfactory readability and inelegant comparison. A great many experiments are conducted, yet tables demonstrating results in this paper are somewhat dispersed in pieces and hard to follow. Different problems/scales/settings own different compared components/methods and reporting manners. A  unified format with as many commonly comparable and explicitly categorized methods is supposed better-organized. Names like AM-bs1024 require explanation. Introduction or classification of methods like SO-mixed is missing. The meaning of the asterisk mark is unclear until Appendix D. Minor points. E.g., in line 167, the second $x_{1,0}$ is likely to be $x_1$? Furthermore, results of T2T are not reported.

**Questions:**

See weakness

---

> ### Author Rebuttal · Authors · 2024-08-03
>
> Thank you very much for your time and effort in reviewing our work. We are glad to know that you find the proposed UDC framework is designed with applicability to a wider range of CO problems, and its performance is shown to be generally good on certain problems.
> >**Weakness 1. Novelty of UDC.**
>
> Thanks for your insightful comments. Neural-divide-and-conquer methods are indeed a combination of existing methods in terms of network structure and solving process. However, we would like to declare that the significant novelty of UDC lies in its **training approach**.
>
> Considering the **training approach**, we do not agree that the technique in the Reunion step resembles prevalent tricks in the neural CO literature. The training methods you mentioned (i.e., POMO/Sym-NCO, etc.) are all for obtaining the RL baseline for single-stage constructive solvers [1]. However, the DCR training method proposed in this article is designed to handle the negative impact of sub-optimal dividing results in neural divide-and-conquer methods (as shown in Table 1 in the original manuscript), which is novel in NCO methods.
>
> As demonstrated in the paper, the DCR training method can significantly facilitate the applicability and effectiveness of UDC.
> >**Weakness 2. Training process.**
>
> Thanks for your suggestion. UDC employs a neural network for the dividing policy and another network for the conquering policy. In a Divide-Conquer-Reunion (DCR) enabled training process, the model for conquering policy will successively update its parameters twice according to the loss $\mathcal{L} _ {c1}(\mathcal{G})$ and $\mathcal{L} _ {c2}(\mathcal{G})$ after the Conquer step and the Reunion step, respectively. Finally, we will calculate the dividing loss function $\mathcal{L} _ {d}(\mathcal{G})$ based on the solution $\boldsymbol{x}_ 2$ and the dividing policy will update the model parameters according to the dividing loss.
>
> A pseudo-code of the training process is provided in Algorithm 1 in Appendix C.6. Please refer to ``Common Concern 2`` for MDP \& Training objectives and we will provide more implementation details to increase the clarity of illustrations.
> >**Weakness 3. Solving time.**
>
> Thanks for your constructive comments. For a better discussion, we respond to the two concerns in Weakness 3 in reverse order.
>
> **The number of conquering stages**: Both the solving time and solution quality of UDC are related to the number of initial solutions $\alpha$ and the number of conquering stages $r$. Therefore, we conduct ablation experiments on the two factors in Appendix E.1 and Appendix E.2, respectively, in the original paper. Changing the $r$ value to adjust solving times, the results in ``Figure 5 of the original paper`` demonstrate that UDC has an efficiency advantage, as its variants generally need less time for better performance compared to the SOTA neural divide-and-conquer methods GLOP and SL-based sub-path solver LEHD.
>
> **Solving time of UDC**: Based on the results above, together with Table 3 and Table 17 of the original manuscript, when generating solutions with $\alpha$=1, UDC can generate acceptable solutions with a significant advantage in solving speed.
>
> Additionally, we are sorry for an index mistake in line 816 where ``Figure 6`` should be ``Figure 5``.
> >**Weakness 4. Supervised learning.**
>
> Although SL-based approaches achieve better performance in some CO problems like TSP and CVRP, we believe that the SL method is not suitable for sub-problem conquering for **Poor applicability**: using SL for sub-problem conquering requires a large number of high-quality heuristic algorithm solutions as training labels. However, in most CO problems, heuristic algorithms are not outstanding (e.g., PCTSP) or are too time-consuming (e.g., min-max mTSP). Therefore, SL-based sub-problem conquering will compromise the availability of UDC for these CO problems.
> >**Weakness 5. Applicability.**
>
> Yes, according to the discussion in Section 3.3, UDC can handle combinatorial optimization problems that meet the conditions.
>
> This paper chooses the 10 CO problems to evaluate UDC, mainly because they have **established datasets, baseline methods, and constructive solvers**. When a new CO problem has developed established datasets, baseline methods, and constructive solvers, it can also be incorporated to evaluate the proposed UDC framework.
> >**Weakness 6. Initial solution.**
>
> Thanks for your valuable suggestion. We further conduct experiments on TSP with different initial solutions, and the results are shown in ``Table 4 of the one-page PDF``. Results demonstrate the contribution of a good initial solution to the final solution. The variants with a random or nearest greedy initial solution cannot converge to good objective values even with 50 UDC conquering stages. Moreover, from TSP500 to TSP2,000, the dividing policy of UDC produces similar results to the heuristic algorithm random insertion, which verifies the quality of the dividing policy.
> >**Weakness 7. Readability.**
>
> Thank you for your suggestion. We will unify the reporting methods, add footnotes, clarify all method names and references, and provide more instructions for appendices accordingly to improve readability. In Table 2 in the original manuscript, we will supplement the reported results of T2T [4].
>
> ***
> >**References**
>
> [1] Berto, Federico, et al. "Rl4co: an extensive reinforcement learning for combinatorial optimization benchmark." arXiv preprint, 2023.
>
> [2] Drakulic, Darko, et al. "Bq-nco: Bisimulation quotienting for efficient neural combinatorial optimization." Advances in Neural Information Processing Systems 36, 2024.
>
> [3] Bertazzi, L. et al. Min–max vs. min–sum vehicle routing: A worst-case analysis. European Journal of Operational Research, 2015.
>
> [4] Yang Li, Jinpei Guo, Runzhong Wang, and Junchi Yan. From distribution learning in training to gradient search in testing for combinatorial optimization. Advances in Neural Information Processing Systems, 36, 2024.

---

> > ### Comment · Reviewer_kAJo · 2024-08-10
> >
> > Thanks for you rebuttal. I would like to raise my scores.

---

> > > ### Author Response · Authors · 2024-08-10
> > >
> > > Thank you for raising the score.

---

### Official Review · Reviewer_h7Dg · 2024-07-08

**Soundness:** 2
**Presentation:** 1
**Contribution:** 2
**Rating:** 5
**Confidence:** 4

**Summary:**

The paper proposes a Divide-Concur-Reunion training approach for solving multiple "large" scale COPs.

**Strengths:**

1- Handling several problems under the same framework with possible different components per CO problem.

2- When compared to learning-based methods, in terms of testing time and solutions sizes, the proposed approach achieves competitive results.

3- The Divide-Concur-Reunion training algorithm.

**Weaknesses:**

[Major Comments]

1- Large- and small-scale instances are not properly defined. Large-scale cannot be only larger instances from previously tested relatively smaller instances by learning-based methods. Scalability is the main bottleneck in many COPs. For example, ILPs are very efficient for smaller scale instances for the MIS problem and are also efficient in larger scale sparse graphs.

2- Furthermore, any learning-based CO solver (or NCO methods) must undergo generalization testing/analysis for every problem. While proposing a method to solve many CO problems is the main goal of the paper, investigating the generalization can differ significantly for each COP. This leads to improved understanding of the limitations/capabilities of the proposed framework. For TSP, the weights and size of the graph are the two parameters to consider. However, for MIS, there are additional parameters such as the graph density and degree distribution. The authors only consider ER700 with p=0.15 (probability of edge creation per node).

3- The proposed solver should tackle what heuristics and exact solvers cannot handle. Comparing run-time results with heuristics is not valid as learning-based methods require datasets and training time (which is ~10 days for the proposed method). For example, LKH3 results of Table 8 indicate that this heuristic requires only 6 seconds (unclear if it is average of total time) to solve 16 instances of OVRP2000 (which is considered large-scale by the authors) whereas the proposed approach takes 20 minutes, excluding the training time.

4- The availability of datasets is a challenging problem. For example, if at testing time, a real-world graph from the SNAP dataset (https://snap.stanford.edu/data/) is presented to UDC. How is it handled? The SNAP graphs do not follow a generative function in NetworkX which means training instances are not available. A discussion is needed here.

5- No details about graph sparsification or why it is needed. In Section 3.1, it is briefly described for TSP and MIS. How about other problems? The paper considers 10 problems. Furthermore, if the edges of the original graph in MIS are maintained, how is that graph sparsification?

6- In the second paragraph of Section 3.1, the write-up covers VRP and TSP, but no discussion is given of the other problems.

7- For the previous two points, Appendix B covers some of the details, but not all. For example, how is the sub-problem preparation done for the MIS?

8- In line 150, if the dividing and conquering stages can be defined as MDP, can you define the MDP explicitly? For example, what are the States set, Transition function, and Reward function?

9- What is “i” in \Omega_i in (5)? If it is from the set defined in line 132, shouldn't it be k?

10- What is the objective function(s) for training the DCR? Since training is the main contribution of the paper, this needs to be explained properly. Saying that "we use the REINFORCE algorithm" is not clear! Algorithm 1 outlines the procedure but does not explain this point. I think the training Algorithm should be placed in the main body of the paper while the gradients in (6) should be placed in the Appendix.

11- What is the functions Rank and Top-k in line 190? Citation(s) or brief description is needed.

12- How are the constraints in Appendix B being handled?

13- Generally, there are many unclear descriptions of the proposed methodology. Even if the proposed approach consists of many components, they need to be properly described and defined. Examples include points 5, 6, 7, 8, 10, 11, and 12.

14- The last three rows of the sub tables in Table 7 (Appendix D) indicate that the proposed approach is not fully unified across different COPs. They do not only suggest that different components are being enabled or disabled for every COP, but also the use of different strategies/architectures of obtaining the policies. For example, the DCR (which is the main contribution of the paper) in the KP problem is disabled.

[Minor]

There are many writing issues and typos in the paper. Few examples are:

1- "exp" in (3) and (4) are denoted using different fonts.

2- Subscripts of \script H is not defined in (4).

3- The title of Section 3.3 can be "Applicability in General CO Problems".

4- "Experiments" in line 199.

5- "Definitions" in line 539.

**Questions:**

See Weaknesses.

**Limitations:**

See Weaknesses.

---

> ### Author Rebuttal · Authors · 2024-08-04
>
> Thank you very much for your time and effort in reviewing our work. We are glad to know that you find the proposed UDC achieves competitive results and can handle several problems under the same framework with possible different components per CO problem.
>
> We sincerely appreciate the effort you have put into reviewing this article and we address your concerns as follows.
> >**Weakness 1. Definition of ``large-scale``.**
>
> We agree that the definition of large-scale cannot only consider the size of instances. As the definition in [1], the large-scale CO problem has more data or decision variables or constraints (large-size sparse graphs might have relatively more decision variables, data, and fewer constraints). We will provide a clearer definition of large-scale CO problems.
> >**Weakness 2 & 4. Generalization.**
>
> We deeply agree that the generalization performance on instances with different parameters is essential to test NCO methods. However, we have to declare that for some CO problems, there are no established datasets to evaluate the generalization. When generalizing the model trained on the ER graph with p=0.15 to the ``Social circles: Facebook`` graph in the SNAP dataset, we find UDC can also output legal solutions with seemingly good objective values but there is no ground truth of MIS on the graph so we cannot check the ability of UDC on it.
>
> We evaluate the generalization performance of UDC on established out-of-domain datasets, e.g., vehicle routing problem (VRP) instances with coordinates sampled from non-uniform distributions. As exhibited in Table 13 in Appendix D.3, the generalization ability of UDC on TSP and CVRP can surpass existing neural solvers. Such advantage may be attributed to the normalization process in the conquering stage.
> >**Weakness 3. Solving time.**
>
> Firstly, we apologize for the typo in the time results of OVRP and we will revise it to the content shown in ``Table 3 of the one-page PDF``, where LKH3 takes 16 minutes for the OVRP500 dataset, 32 minutes for OVRP1,000, and 27 minutes for OVRP2,000.
>
> We agree that neural methods should tackle what heuristics and exact solvers cannot handle, especially when heuristics has only low performance or requires an unacceptable time to generate solutions. In certain CO problems, UDC can compensate for the above two situations. Considering **performances**, UDC can produce better results than all available existing heuristics in (S)PCTSP and min-max mTSP. For **solving time**, we acknowledge that NCO methods require additional training time, which is time-consuming. However, well-trained models, such as UDC for large-scale OP and TSP, can generate good solutions within several seconds. Compared to heuristics, the online solving characteristic of UDC may be valuable for practical applications that require response speed.
> >**Weakness 5-7, 13. Pipeline settings.**
>
> Thanks for your comments. We will improve the expression in Section 3 and Appendix B based on your suggestions to eliminate ambiguity, and we will also publish all the code after the paper is published to show the detailed implementation.
>
> For weakness 5, in non-graph input CO problems, especially VRPs, constructing sparse graphs aims at deleting more useless edges to reduce the time complexity. It is a general step for existing heatmap-based solvers.  By processing only sparse graphs, each layer of GNNs can achieve a time complexity of O ($KN$) ($K$ is the neighbor size, usually 100 in UDC), which is helpful to large-scale VRPs. In addition, this procedure is to construct sparse graphs for instances rather than ``graph sparsification``. For MIS, the ER graph with p=0.15 is a sparse graph itself so there are no additional steps needed for constructing sparse graphs. For weakness 7, Sub-MIS is generated with random nodes from the original MIS.
>
> >**Weakness 8 & 10. MDP & training.**
>
> Please refer to ``Common Concern 2`` for the MDP definition & objective functions.
>
> >**Weakness 11: Line 190.**
>
> Only CO problems with decomposable aggregate functions [2] can be solved by the divide-and-conquer method. So UDC can not apply to CO problems with indecomposable aggregate functions such as minimizing the visiting rank of given nodes or the top-k longest route in a graph.
>
> >**Weakness 12. Constraint handling.**
>
> Feasibility masks will mask the illegal action in generating both initial solutions and sub-solutions (You can also refer to ``Question 1 of reviewer Eauz`` for details).
>
>
> >**Weakness 14. Unified configuration.**
>
> We need to declare that 'unified' in this article means that UDC adopts a unified training framework for the diving and conquering stages. For the 10 CO problems involved, there is no unified conquering policy that can solve all problems (Please refer to ``Common Concern 1``) so UDC adopts different conquering policies for different CO problems.
>
> **The third to last** and **the second to last** rows of Table 7 are related to the environment or constraint of different CO problems. For KP, the sub-optimal connection shown in Figure 2 in the original paper will not occur, so there is no optimality difference in whether DCR is enabled. We provide the KP results with DCR enabled in ``Table 2 of the one-page PDF`` and these results support the above conclusion.
>
> The two-side conquering baseline may not be applicable in certain CO problems like ATSP and CVRP where reversing the sub-solution will be illegal, so we disable such baseline in these problems. We will add an explanation to summarize all the settings in Table 7.
>
> >**Minor Weakness & Weakness 9. Typos.**
>
> Yes, the $\Omega_i$ should be $\Omega_k$. Thanks for pointing typos out, we will fix all of them accordingly.
>
> ***
> >**References**
>
> [1] Gervet, Carmen. "Large scale combinatorial optimization: A methodological viewpoint." DIMACS series, 2001.
>
> [2] Jesus, Paulo, Carlos Baquero, and Paulo Sérgio Almeida. "A survey of distributed data aggregation algorithms." IEEE Communications Surveys & Tutorials, 2014

---

> ### Comment · Reviewer_h7Dg · 2024-08-09
> **My concerns are partially addressed.**
>
> Definition of large Scale instance:
>
> - CO problems have been well-studied, with a history dating back to the seventies (see "Reducibility among combinatorial problems" for an example). Understanding the difficulty of each problem should give the authors a better sense of which instances are more challenging to solve and, therefore, evaluate them accordingly. Harder instances may not necessary be larger instances. For the MIS example, mid-density graphs (w.r.t. complete graphs) with n~2000 is much harder to solve in KaMIS and ILP solvers when compared to the very-large and very sparse SNAP dataset graphs (with million of nodes).
>
> Generalization:
>
> - Generalization in COPs shouldn't rely on the availability of a dataset. Most datasets in COPs were specifically created for ML4CO methods, but that doesn't mean datasets will always be available in practice. For example, in the case of MIS, apart from the SATLIB dataset (which represents hard, sparse instances compared to ER), there are no other datasets with accurately known MISs.
>
> - The generalization test can be as simple as the following: Evaluate your ER-trained model with different values of p and n. Does your model require different graph embeddings for larger graph sizes?
>
> - For SNAP, you can compare with SOTA heuristics (KaMIS) unless the proposed model requires different embeddings or needs to be trained from scratch (or possibly fine-tuned). If any of these conditions apply, they should be stated and discussed.
>
> - Prior to the development of DNNs, we didn't know how to classify images. This is not the case for COPs as they are not typical supervised ML problems. For COPs, we already have (1) well-performing problem-specific heuristics and/or ILP solvers, and (2) a large body of previous problem-specific studies that investigate the hardness of every problem.
>
> - Please note that I am not saying that the proposed method should obtain SOTA results (in terms of both solutions sizes and run-time) on all CO problems on all instances, but I am saying that the proposed framework should be evaluated such that the limitation/capabilities are clear and understood with the main target: Can the proposed framework, by any metric, outperform SOTA problem-heuristics that does not require any data on a specific instance(s)? For MIS, the results in Table 9 says otherwise if we count for the nearly 10 days of training. Even if we exclude the training time and observe only that the inference time (21.05m), can the proposed method outperform KaMIS when p or n change for ER? These are the types of questions that should be accounted when evaluating the 10 problems.
>
> - I acknowledge the authors results for TSP in Tables 3 and 4 of the one-page PDF in the rebuttal. This is a good example of evaluating the performance UDC for a CO problem.
>
>
> Overall:
> - My comments on solving time, pipeline Settings, description of MDP and the training algorithm, and constraints handling are well explained by authors. I highly recommend to integrate these into subsequent versions of the paper.
> - I think the advantage of this proposed framework is obtaining solutions (not necessarily the best in terms of solution quality and run-time) to some instances of 10 different CO problems.
> - I thank the authors for their efforts and the additional experiments. As such, I will increase my score to 5 as I still believe that the capabilities and limitations of the proposed framework of most problems are not well-understood.

---

> > ### Author Response · Authors · 2024-08-10
> >
> > Thank you very much for your valuable comments and raising your score to 5. We are glad to hear that some of your concerns have been resolved and that you recognize the advantage of UDC in applicability. We will include these details in our manuscript and provide a formal definition of large-scale accordingly.
> >
> > Thank you very much for your helpful suggestions on the generalization study. We will try our best to accomplish these experiments as soon as possible and provide the results before the end of the rebuttal period.
> >
> > By the way, the score in the system seems to remain unchanged. Could you please double-check it?

---

> > > ### Comment · Reviewer_h7Dg · 2024-08-10
> > > **Comment about generalization with MIS**
> > >
> > > Could the authors answer the question I raised in the 2nd point of Generalization.
> > >
> > > In regard to the score, I did not see the option to change the score.

---

> > > > ### Author Response · Authors · 2024-08-10
> > > >
> > > > Thank you very much for your prompt reply. We will conduct the experiment mentioned in the 2nd point of Generalization first.
> > > >
> > > > The option to change the score can be found by editing the initial submitted official review.

---

> > > > > ### Comment · Reviewer_h7Dg · 2024-08-10
> > > > > **Thank you**
> > > > >
> > > > > Thanks for pointing this out. I did change the score.
> > > > >
> > > > > My question can be answered without an experiment. I wanted to know if the proposed model, when tested with graphs with larger size than the training set, does the encoding change? If yes, would newly un-optimized parameters be fine-tuned or left randomly chosen?

---

> ### Author Response · Authors · 2024-08-10
>
> Thank you for changing the score. Sorry, we're a little confused about your question. Does the graph embedding refer to the embedding itself or its size?
>
> If it is the former, different graph inputs are bound to get different embeddings.
>
> If it is the latter, the answer is no. The calculation methods of embeddings are unchanged regardless of the size of the input graph. The obtained graph embeddings on each vertex are always d-dimensional vector $\mathcal{R}^{d}$ and no new parameters will be introduced.
>
> When AGNN computes the graph embedding of each vertex, it only needs the neighborhood information of this vertex, without the full graph information. Therefore the computation of embeddings is independent of the graph size.

---

> ### Author Response · Authors · 2024-08-12
> **Further experiments to validate the capabilities and limitations.**
>
> >**Further experiments to validate the capabilities and limitations**:
>
> Thank you for your valuable suggestions on generalization experiments. In addition to the TSP and CVRP, we conduct generalization experiments on the MIS problems. We conduct zero-shot generalization on the model trained by ER graphs with n$\sim U(700,800)$ and p=0.15 to five other datasets, including:
>
> * A random ER graph dataset with n=720 and p=0.05. (average 12,960 undirected edges)
>
> * A random ER graph dataset with n=720 and p=0.25. (average 64,800 undirected edges)
>
> * A random ER graph dataset with n=2,000 and p=0.05. (average 100,000 undirected edges)
>
> * The ego-Facebook data in SNAP. (4,039 nodes, 88,234 undirected edges)
>
> * The feather-lastfm-social data in SNAP. (7,624 nodes, 27,806 undirected edges)
>
> All the first three ER random datasets contain 128 graph instances, and we consider the last two data as a single non-connected graph. The table below exhibits the comparison results between UDC and the SOTA heuristics KaMIS (implemented on https://github.com/KarlsruheMIS/KaMIS) on the original in-domain dataset (ER-[700-800], p=0.15) and the five out-of-domain datasets mentioned above. Obj. represents the objective function value, Gap represents the performance gap compared to the best algorithm and Time is the time consumption for solving the whole dataset.
>
>
> | Dataset                   || |ER, n=720,   p=0.05 | |||ER-[700-800], p=0.15| (in-domain)       |
> |---------------------------|-|-------------|-------------|---------|-|-----------------|-----------------|-----------------|
> | Method                   | | Obj.        | Gap         | Time   | | Obj.            | Gap             | Time            |
> | KaMIS                    | | 99.5        | -           | 40.6m  | | 44.9            | -               | 52.13m          |
> | UDC-$x_0$($\alpha=50$)    || 79.5        | 20.10%      | 8s   |   | 41.0            | 8.62%           | 40s             |
> | UDC-$x_{50}$($\alpha=50$) || 89.5        | 10.07%      | 9.03m  | | 42.9            | 4.44%           | 21.05m          |
> | UDC-$x_{250}$($\alpha=50$)|| 94.5        | 5.03%       | 44.84m  || 43.8            | 2.41%           | 1.73h           |
> | **Dataset**               | ||**ER, n=720, p=0.25**   |||| **ER, n=2,000, p=0.05**               |   |
> | Method                   | | Obj.        | Gap         | Time  |  | Obj.            | Gap             | Time            |
> | KaMIS                    | | 28.2        | -           | 1.4h   | | 133.9           | -               | 3h              |
> | UDC-$x_0$($\alpha=20$)    || 21.6        | 23.46%      | 58s   |  | 116.9           | 12.72%          | 2m              |
> | UDC-$x_{50}$($\alpha=20$)| | 25.3        | 10.54%      | 21m  |   | 119.1           | 11.05%          | 21m             |
> | UDC-$x_{250}$($\alpha=20$)|| 26.4        | 6.59%       | 63m   |  | 122.9           | 8.25%           | 90m             |
> | **Dataset**                 | |**SNAP-** | **ego-Facebook** | ||**SNAP-**| **feather-lastfm-social**            ||
> | Method                   | | Obj.        | Gap         | Time  |  | Obj.            | Gap             | Time            |
> | KaMIS                    | | 1052.0      | -           | 155s  |  | 4177.0          | -               | 0.1s            |
> | UDC-$x_0$($\alpha=1$)     || 767.0       | 27.09%      | 6s |     | 3622.0          | 13.29%          | 2s              |
> | UDC-$x_{250}$($\alpha=1$)| | 901.0       | 14.35%      | 11s|     | 3751.0          | 10.20%          | 13s             |
> | UDC-$x_{2500}$($\alpha=1$)|| 1009.0      | 4.09%       | 60s   |  | 4067.0          | 2.63%           | 80s             |
>
>
> Compared to KaMIS, UDC variants exhibit relatively stable performance gaps across datasets with various node sizes (n), sparsity (p), and generation methods, demonstrating good generalization ability. Moreover, except for the extremely sparse graph SNAP-feather-lastfm-social, UDC variants have significant advantages over KaMIS in terms of time.
>
>
> As for learning-based methods, currently, we are not able to run the given code of any learning-based algorithm with given instruction, so we are temporarily unable to compare the generalization ability with other learning-based algorithms. We will try to include these learning-based baselines in future manuscripts. In addition, besides MIS, we will also strive to conduct generalization tests on more CO problems.
>
> We hope the above discussion could address your remaining concerns. Please let us know if you have any further concerns.

---

### Official Review · Reviewer_Eauz · 2024-07-12

**Soundness:** 3
**Presentation:** 3
**Contribution:** 3
**Rating:** 6
**Confidence:** 3

**Summary:**

The paper introduces a Unified Neural Divide-and-Conquer (UDC) framework designed to tackle large-scale combinatorial optimization problems by leveraging a novel training methodology called Divide-Conquer-Reunion (DCR). This framework employs graph neural networks for the division of problems and utilizes established constructive solvers for conquering sub-problems, aiming to optimize the entire process. UDC has been evaluated across a range of benchmark datasets and demonstrated superior performance compared to existing neural and heuristic methods, particularly in scalability and solution quality.

**Strengths:**

1. **Innovative Framework**: UDC integrates a novel training methodology that effectively addresses the challenge of sub-optimal divide policies, improving overall solution quality.
2. **Extensive Applicability**: The framework demonstrates broad applicability across various large-scale combinatorial problems, confirming its versatility and effectiveness.
3. **Strong Empirical Results**: UDC outperforms several baseline methods on a diverse set of problems, which strongly supports the utility and efficiency of the approach.

**Weaknesses:**

**Dependence on Specific Architectures**: The success of the method heavily relies on the choice of graph neural networks and the specific configurations of constructive solvers, which might not generalize across all types of combinatorial optimization problems.

**Questions:**

1. How does the UDC framework ensure that feasible initial solutions are generated during the division stage, particularly when traditional heuristics are not applicable?
2. Does the UDC framework require extensive expert intervention for tuning neural network parameters and training regimens to achieve optimal performance, especially when adapting to new or unseen problem types?

**Limitations:**

**Constraint Handling**: UDC may not perform well on problems with intricate constraints (e.g., time windows in routing problems) that require more than just feasibility checks.

This is an interesting field, and if the authors can provide sufficiently detailed responses to the questions raised, I would be willing to consider revising my score.

---

> ### Author Rebuttal · Authors · 2024-08-05
>
> Thank you very much for your time and effort in reviewing our work. We are glad to know that you find the proposed UDC integrates a novel training methodology, demonstrates broad applicability across various large-scale CO problems, and outperforms several baseline methods.
>
> We address your concerns as follows.
>
> >**Weakness 1. Dependence on specific architectures:** The success of the method heavily relies on the choice of graph neural networks and the specific configurations of constructive solvers, which might not generalize across all types of combinatorial optimization problems.
>
> Thanks for your insightful comments. However, we respectfully disagree with you that the success of UDC heavily relies on the choice of graph neural networks and the specific configurations of constructive solvers. Our reasons are as follows:
>
> **Constructive solvers**: As illustrated in ``Common Concern 1`` of the general response, UDC involves 10 CO problems and it is impossible to employ the same constructive solver for all CO problems [1]. However, UDC exhibits outstanding flexibility in employing different constructive solvers as the conquering policy. Experiments mentioned in ``Common Concern 1`` show that the final performance of UDC does not rely on a specific constructive model for the conquering policy, which means that any available constructive solver with good performance can be chosen.
>
> **Graph neural networks (GNNs)**: The current UDC framework only evaluates the AGNN for the dividing policy across all 10 CO problems. However, the use of AGNN [2] in UDC is not a specific choice as well. **1)** UDC adopts GNN instead of other network structures because of its lightweight memory and time consumption and **2)** AGNN is one of the most basic forms of massage-passing neural network [3], which only calculates the bidirectional edge information and bidirectional node information within neighbors. **3)** When using AGNN to process embeddings of all 10 CO problems, UDC follows a consistent configuration.
>
>
> >**Question 1. feasible initial solution**:  How does the UDC framework ensure that feasible initial solutions are generated during the division stage, particularly when traditional heuristics are not applicable?
>
> For involved CO problems, UDC autoregressively constructs the initial solution node-by-node based on the heatmap generated by GNN. At every step, it employs a **feasibility mask mechanism** to ensure the feasibility of the final generated solution. This mechanism is fundamental in both heatmap-based solvers [4] and constructive solvers [5]. It prevents the selection of nodes that would lead the next partial solution outside the set of all feasible solutions $\Omega$. For example, the feasibility mask mechanism for TSP will mask the selection probabilities of all visited nodes to 0.
>
> With the feasibility mask, the dividing policy can construct valid solutions for most CO problems. UDC cannot apply to the relatively rare CO problems where the feasibility mask mechanism cannot ensure a valid initial solution.
>
> >**Question 2. Requirement on tunning**: Does the UDC framework require extensive expert intervention for tuning neural network parameters and training regimens to achieve optimal performance, especially when adapting to new or unseen problem types?
>
> UDC does not conduct special tuning for specific CO problems. As shown in Appendix D.1, we use the same learning rate and AGNN structure across all 10 CO problems.
>
> When applying to a new problem type, UDC requires:
>
> **1)** Employing an available constructive solver for conquering policy (Please refer to ``Common Concern 1`` \& ``Weakness1``).
>
> **2)** Set up new RL environments and feasibility masks based on constraints.
>
> **3)** Changing $\alpha$ and $\beta$ to maximize the CUDA memory usage. Note that this change is not for solving performance and will only facilitate training efficiency.
>
>
> ***
>
> >**References**
>
> [1] Kwon, Yeong-Dae, et al. "Matrix encoding networks for neural combinatorial optimization." Advances in Neural Information Processing Systems 34, 2021.
>
> [2] Xavier Bresson and Thomas Laurent. "An experimental study of neural networks for variable
> graphs." In ICLR 2018 Workshop, 2018.
>
> [3] Gilmer, Justin, et al. "Message passing neural networks." Machine learning meets quantum physics, 2020.
>
> [4] Ruizhong Qiu, Zhiqing Sun, and Yiming Yang. DIMES: A differentiable meta solver for combinatorial optimization problems. In Advances in Neural Information Processing Systems 35, 2022.
>
> [5] Zhou, Changliang, et al. "Instance-Conditioned Adaptation for Large-scale Generalization of Neural Combinatorial Optimization." arXiv preprint, 2024.

---

> > ### Comment · Reviewer_Eauz · 2024-08-11
> > **Response to Rebuttal**
> >
> > Thank you for your detailed and thoughtful rebuttal. Your explanations have addressed my concerns, particularly regarding the flexibility of the UDC framework in terms of architecture and solver selection. I now have a better understanding of how the framework ensures feasible solutions during the division stage and the level of tuning required for different problem types.
> >
> > This is indeed an impressive framework with broad applicability across various combinatorial optimization problems. I am satisfied with your responses and will maintain my positive score.

---

> > > ### Author Response · Authors · 2024-08-11
> > >
> > > Thanks for your support and positive score. We are glad to hear your appreciation of the broad applicability of UDC and that our explanation has addressed your concerns.

---

### Author Rebuttal · Authors · 2024-08-05

We sincerely thank all reviewers for their constructive comments and valuable suggestions. These suggestions greatly help us to improve our manuscript. The current manuscript has received extensive positive evaluations regarding its novelty, effectiveness, applicability, and impact:
* **Reviewer Eauz**: UDC integrates a novel training methodology that effectively addresses the challenge of sub-optimal divide policies. It demonstrates broad applicability across various large-scale combinatorial problems. UDC outperforms several baseline methods on a diverse set of problems, which strongly supports the utility and efficiency of the approach.
* **Reviewer h7Dg**: When compared to learning-based methods, in terms of testing time and solution sizes, the proposed UDC achieves competitive results.
* **Reviewer kAJo**: The UDC framework is designed with applicability to a wider range of CO problems, without heavy reliance on problem-specific heuristics like in the previous works. The authors conduct extensive experiments where the proposed framework's performance is shown to be generally good on certain problems.
* **Reviewer Uqww**: This paper conducts extensive experiments. UDC exhibits effectiveness in 10 different CO problems, improving the existing divide-and-conquer methods in both effectiveness and applicability. The idea of DCR is quite novel and experiments give detailed explanations to the setting reason for hyperparameters and components.

***

We address some common concerns shared by different reviewers in this response.

>**Common Concern 1**: Explanation of choosing different constructive solvers (i.e., models for conquering policy) for different CO problems. (Reviewer Eauz, Reviewer h7Dg)

Using the same model for conquering policy is an ideal choice. However, the experiment of UDC involves 10 CO problems, and some CO problems, like min-max mTSP [1] and ATSP [2], require specific model structures to process their inputs and constraints. Consequently, there is no constructive solver that can be used for all 10 involved CO problems. For each CO problem, UDC adopts the best-performance constructive solver for the conquering policy (i.e., summarized in Appendix Table 7).

UDC exhibits flexibility in choosing the constructive solver for the conquering policy. For well-developed CO problems like TSP and KP, there are multiple constructive solvers available. Table 5 in the original paper presents an ablation study on the selection of these constructive solvers for TSP. In rebuttal, we conduct a similar ablation study on KP, and the results are displayed in ``Table 1 of the appendix one-page PDF``. Both sets of results indicate that the final performance of UDC does not rely on a specific constructive model for the conquering policy.

***

>**Common Concern 2**: Presentation of training objective and their Markov Decision Process (MDP) (Reviewer h7Dg, Reviewer kAJo)


In addition to Algorithm 1 in Appendix C.6, we will include descriptions of the training objective of both policies and the MDP for the two stages.

**Objective functions of optimizing the two networks**: For a CO problem (or a sub-CO problem) with an objective function $f(\cdot)$, the objective of optimizing both networks is to maximize the reward functions of their corresponding MDP. The reward is $-f(\boldsymbol{x}_2)$ for the GNN, $-f(\boldsymbol{s}^1)$ for the constructive solver in the Conquer step (i.e., the first conquering stages), and $-f(\boldsymbol{s}^2)$ in the Reunion step.


**MDP for the dividing stage**: The MDP $\mathcal{M}_d=${$\mathcal{S}_d,\mathcal{A}_d,\boldsymbol{r}_d,\mathcal{P}_d$} of the dividing stage can be represented as follows:

* State. The state ${st} _ d \in \mathcal{S} _ d $ represents the current partial solution. The state in the $t$-th time step is the current partial solution with $t$ nodes ${st} _ {d,t}=\boldsymbol{x} _ {0,t}=(x_{1},x_2,\ldots,x_t)$. $st_{d,0}$ is empty, $st_{d,T}=\boldsymbol{x}_0$.

* Action \& Transaction. The action is to select a node at time step $t$, i.e., $a_{d,t}=x_{t+1}$. The chosen node needs to ensure that the partial solution ${st} _ {d,t+1}=(x _ 1,\ldots,x _ t,x _ {t+1}) $ is valid (i.e., $st _ {d, T}\in \Omega$).

* Reward. Every single time step has the same reward $r_{d,t}=-f(\boldsymbol{x}_2)$, which is the negative objective function value of the final solution $\boldsymbol{x}_2$ after the whole DCR process.

* Policy. The policy $\pi_d$ is shown in Eq. (4) of the original paper.

**MDP for the conquering stage**: The MDP $\mathcal{M}_c=${$\mathcal{S}_c,\mathcal{A}_c,\boldsymbol{r}_c,\mathcal{P}_c$} of any single conquering stage is represented similarly as follows:

* State. Each state ${st} _ c\in \mathcal{S} _ c$ represents a partial solution of sub-CO problems. The state in the $t$-th time step is the current partial (sub-)solution with $t$ nodes ${st} _ {c,t}=\boldsymbol{s} _ {t}=(s_{1},s_2,...,s_t)$. $st_{c,0}$ is empty and $st_{c,T}=\boldsymbol{s}$.

* Action \& Transaction. The action is to select a node at time step $t$ as well, i.e., $a_{c,t}=s_{t+1}$.

* Reward. The reward in each time step becomes the objective value of sub-CO solution $\boldsymbol{s}$, i.e., $r_{c,t}=-f(\boldsymbol{s})$.

* Policy. The policy $\pi_c$ is shown in Eq. (5) of the original paper.


***


Point-to-point responses can be found below. We are also glad to continually improve our work to address any further concerns.

Best Regards,

Paper12083 Authors

***

>**Reference**


[1] Son, Jiwoo, et al. "Equity-Transformer: Solving NP-Hard Min-Max Routing Problems as Sequential Generation with Equity Context." AAAI 2024, 2024.

[2] Kwon, Yeong-Dae, et al. "Matrix encoding networks for neural combinatorial optimization." NeurIPS 2021, 2021.

---

### Author Response · Authors · 2024-08-14
**Message to AC**

Dear Area Chair,

Thank you very much for your effort in organizing the review of our work. Here is our summary of the author-reviewers discussion.

This paper presents a novel neural divide-and-conquer framework termed UDC for large-scale CO problems. UDC proposes a novel method DCR to enhance training by alleviating the negative impact of sub-optimal dividing policies. Leveraging DCR in training, the proposed UDC achieves a unified neural divide-and-conquer training scheme, it demonstrates extensive applicability and has been evaluated on 10 CO problems with outstanding performances.

After the discussion period, **all four reviewers** appreciated the effectiveness and broad applicability of UDC.

**Reviewer Uqww** found the article easy to follow, the idea of DCR is quite novel, and experiments give detailed explanations on the setting reason for hyperparameters and components. Reviewer Uqww raised the rating from Weak Accept (6) to **Accept (7)** after the discussion and believed that our work ``is inspiring to the NCO research``.

**Reviewer Eauz** found that UDC integrates a novel training methodology that effectively addresses the challenge of sub-optimal divide policies and rated our work as **Weak Accept (6)**.

**Reviewer kAJo** found the proposed UDC does not heavily rely on problem-specific heuristics compared to the previous works and the idea of considering interdependencies between dividing and conquering stages is interesting as a neural CO training scheme. Reviewer kAJo raised the rating from Borderline Reject(4) to **Borderline Accept (5)** after the discussion.

**Reviewer h7Dg** found that UDC handles several problems under the same framework with possible different components per CO problem. After discussion, reviewer h7Dg found some of the comments were well explained and raised the rating from Reject(3) to **Borderline Accept (5)**. Reviewer h7Dg further suggested conducting the out-of-domain generalization experiments on CO problems besides TSP and CVRP to validate the capabilities and limitations of UDC. We have supplemented the generalization experiments of UDC on the MIS problem and will also conduct the generalization tests on more CO problems in the camera-ready version accordingly.

Best Regards,

Paper12083 Authors

---

### Decision · Program_Chairs · 2024-09-25

**Decision:**

Accept (poster)

**Comment:**

After extensive discussion, all reviewers concur that this interesting paper, which advances the size and generality of the combinatorial optimization problems that can be solved with a deep-learning approach, should be accepted.